# REAL-TIME ROBOT EXECUTION WITH MASKED ACTION CHUNKING

**Haoxuan Wang**[1,3]**, Gengyu Zhang**[1,3]**, Yan Yan**[1]**,
Yuzhang Shang**[2]**, Ramana Rao Kompella**[3]**, Gaowen Liu**[3,†]
[1]University of Illinois Chicago    [2]University of Central Florida    [3]Cisco Research

## ABSTRACT

Real-time execution is essential for cyber-physical systems such as robots. These systems operate in dynamic real-world environments where even small delays can undermine responsiveness and compromise performance. Asynchronous inference has recently emerged as a system-level paradigm for real-time robot manipulation, enabling the next action chunk to be predicted while the current one is being executed. While this approach achieves real-time responsiveness, naive integration often results in execution failure. Previous methods attributed this failure to *inter-chunk discontinuity* and developed test-time algorithms to smooth chunk boundaries. In contrast, we identify another critical yet overlooked factor: *intra-chunk inconsistency*, where the robot's executed action chunk partially misaligns with its current perception. To address this, we propose REMAC, which learns corrective adjustments on the pretrained policy through masked action chunking, enabling the policy to remain resilient under mismatches between intended actions and actual execution during asynchronous inference. In addition, we introduce a prefix-preserved sampling procedure to reinforce inter-chunk continuity. Overall, our method delivers more reliable policies without incurring additional latency. Extensive experiments in both simulation and real-world settings demonstrate that our method enables faster task execution, maintains robustness across varying delays, and consistently achieves higher completion rates. Video demos are available at this project page.

## 1 INTRODUCTION

The deployment of large-scale models has revolutionized AI capabilities across various domains (Liu et al., 2023; Esser et al., 2024; Zheng et al., 2024; OpenAI, 2025). In robotics, Vision-Language-Action (VLA) models (Wu et al., 2023; Brohan et al., 2023b;a; Kim et al., 2024; Black et al., 2024; Intelligence et al., 2025a; Yang et al., 2025; Zhong et al., 2025) have emerged as a promising direction for translating human instructions and sensory observations into physical actions. While *real-time* responsiveness is desirable across all such applications, its importance varies by context. For language or image models, slower responses primarily lead to increased waiting time but won't result in generation failure. In contrast, for cyber-physical systems like robots operating in dynamic, real-world environments, the absence of real-time responsiveness can mean the difference between successfully filling a cup and catastrophically spilling juice onto the table.

A real-time robot control system must ensure a continuous stream of executable actions so that the robot never runs idle (Hester et al., 2011). A straightforward approach is to reduce the time cost of action acquisition—primarily arising from VLA action generation and network communication—such that it falls below the robot's control period. Although a variety of model-level techniques have been proposed to accelerate model prediction (Leviathan et al., 2023; Bolya et al., 2023; Wu et al., 2024; Lin et al., 2024; Shang et al., 2024; Xu et al., 2024; Shukor et al., 2025), these methods typically sacrifice accuracy for speed. Moreover, as model sizes continue to scale, control frequencies increase, and deployment conditions vary in real-world settings, such approaches become increasingly limited in their ability to guarantee real-time performance.

Action chunking (Zhao et al., 2023; Chi et al., 2024), where a policy predicts and executes a sequence of actions per inference step, offers a partial solution to real-time control. Although it has

achieved state-of-the-art results in dexterous manipulation, action chunking inherently reduces system reactivity to sudden state changes (Liu et al., 2025) and introduces discontinuities at chunk boundaries (Black et al., 2025). These limitations have motivated the search for more general system-level inference frameworks, among which ***asynchronous inference*** (Shukor et al., 2025) has recently emerged as a promising solution for real-time robotic execution. Unlike synchronous inference (Chi et al., 2024), which pauses until future actions are available, asynchronous inference predicts upcoming actions while executing the current ones, ensuring a continuous supply of actions. However, incorporating asynchronous inference into VLA-based control systems is nontrivial: when coupled with action chunking, it amplifies existing limitations and results in substantial performance degradation (Liu et al., 2025; Black et al., 2025; Shukor et al., 2025).

Previous methods (Zhao et al., 2023; Liu et al., 2025; Black et al., 2025) addressing this challenge primarily aimed to mitigate **inter-chunk discontinuity**, seeking a balance between long-term coherence and short-term reactivity. These approaches typically treat the already executed actions as informative priors and apply test-time refinements to the upcoming action chunks. However, such refinements are either heuristic in nature and prone to catastrophic failure (Zhao et al., 2023), or exploit the generative and inpainting capabilities of diffusion/flow-matching models (Song et al., 2023) while incurring additional latency (Liu et al., 2025; Black et al., 2025). Moreover, these works overlook a critical failure mode: **intra-chunk inconsistency**, arising from misalignment between observations and the actions executed within a single chunk.

In this work, we aim to improve both inter-chunk continuity and intra-chunk consistency when integrating asynchronous inference with chunking policies. We first formulate intra-chunk inconsistency as a partial mismatch between observations and executed actions, leading to a shift in intra-chunk distribution between training and sampling. To address this, we propose **R**eal-time **E**xecution with **M**asked **A**ction **C**hunking (REMAC) to learn corrective adjustments to the pretrained policy by masking arbitrary portions of action chunks. In parallel, we enhance inter-chunk continuity by refining the sampling pipeline to incorporate previously executed actions as informative priors.

Our method introduces no additional inference delay compared to the pretrained policy (Black et al., 2024; 2025), and can be seamlessly integrated into existing VLA frameworks. Extensive experiments across 12 simulated tasks and three real-world settings demonstrate its effectiveness, showing higher success rates, faster task completion, and smoother robot dynamics under varying delay conditions. Moreover, our approach can be combined with existing test-time algorithms, further underscoring its utility to produce stronger backbone policies for asynchronous execution.

## 2 RELATED WORKS

### 2.1 VLA AND ACTION CHUNKING

An emerging line of robotics research focuses on developing generalist policies capable of performing a broad spectrum of tasks and transfer across different environments and robot embodiments. Vision-Language-Action (VLA) models (Brohan et al., 2023b;a; Wu et al., 2023; Kim et al., 2024; Black et al., 2024; Bjorck et al., 2025; Intelligence et al., 2025a; Shukor et al., 2025) have become a leading approach in this pursuit. By conditioning on high-level inputs such as video frames, natural language instructions, and proprioceptive signals, VLAs generate the corresponding low-level motor control commands. Among these models, diffusion and flow-matching approaches (Lipman et al., 2023; Chi et al., 2024) have gained prominence for their ability to generate high-quality actions efficiently, and they serve as the representative setting for our work.

Moreover, scaling to long-horizon manipulation requires temporal abstraction: models must predict not just the next command, but coherent segments of behavior. Inspired by principles of human motor control (Lai et al., 2022), recent systems structure behavior into temporally extended sequences, or *action chunks*, which are generated by the VLA and executed by a low-level controller. This action-chunking paradigm enables visuomotor policies to operate over meaningful sequences and provides a foundation for alternative strategies of action execution.

Figure 1: Illustration of execution paradigms. Arrowed lines of the same style indicate processes occurring simultaneously. **(a)** Synchronous inference: VLA prediction and robot execution alternate sequentially. **(b)** Asynchronous inference: VLA prediction runs concurrently with execution. **(c)** Although asynchronous inference enables real-time execution, it introduces two performance-degrading challenges: exacerbated inter-chunk discontinuity and intra-chunk inconsistency.

## 2.2 EXECUTION STRATEGIES FOR CHUNKED POLICIES

While action chunks are temporally consistent within a single segment, discontinuities and distribution shifts often arise at chunk boundaries. To mitigate this issue, Zhao et al. (2023) proposed Temporal Ensembling (TE), which aggregates the overlapping portions of consecutive chunks to improve smoothness. Liu et al. (2025) highlighted the importance of balancing long-term consistency with short-term reactivity, introducing Bidirectional Decoding (BID), a method that samples multiple candidate predictions and selects the optimal one. More recently, RTC (Black et al., 2025) explored real-time execution strategies under asynchronous inference by framing the task as a *test-time* inpainting (Pokle et al., 2024) problem, proposing to leverage the prior chunk to warm-start planning for the next and applies gradient-based corrections to the predicted chunk. However, RTC introduces additional computational latency, which may degrade performance. In this paper, we likewise target real-time execution under asynchronous inference, but instead adopt a *training-time* adaptation approach. Our method introduces no additional inference overhead, consistently outperforms existing strategies, and can be seamlessly combined with test-time correction methods.

## 3 PRELIMINARIES

Let $\mathbf{v}_\pi(\mathbf{A}_t|\mathbf{o}_t)$ denote the action chunking policy, where $\mathbf{A}_t = [\mathbf{a}_t, \mathbf{a}_{t+1}, \ldots, \mathbf{a}_{t+P-1}]$ represents the predicted action chunk of length $P$, $\mathbf{o}_t$ denotes the current observation, and $t$ indexes the controller timestep. The parameter $P$ is the *prediction horizon*. During rollout, only the first $h$ actions are executed, where $h$ is the *execution horizon* with $1 \leq h \leq P$. In real-world robotic manipulation, latency arises from stages such as VLA action prediction and network communication, which together define the *inference delay*: a continuous quantity representing the time lag between the acquisition of $\mathbf{o}_t$ and the availability of the corresponding $\mathbf{A}_t$ in the control queue. Following Black et al. (2025), we discretize this delay as $d := \lfloor \delta/\Delta t \rfloor$, where $\delta$ is the continuous inference latency and $\Delta t$ is the high-level controller sampling period. For simplicity, our definition of inference delay excludes observation latency and sub–timestep–level delays.

*Flow-matching policies.* We follow prior work (Liu et al., 2025; Black et al., 2025) and consider action-chunking policies trained via flow matching (Lipman et al., 2023). In standard flow matching, the model learns a velocity field that maps intermediate action states toward expert trajectories. Training proceeds by minimizing an $\ell_2$ objective between the predicted flow $\hat{\mathbf{u}}$ and the ground-truth target $\mathbf{u}$. At inference time, a flow-matching policy generates an action chunk by first sampling an initial latent action sequence $\mathbf{A}_t^0$ from a Gaussian prior. The final chunk is then obtained by integrating this sample along the learned velocity field $\mathbf{v}_\pi$ over a normalized time variable $\tau \in [0, 1]$. Using $n$ integration steps, the update rule is:

$$\mathbf{A}_t^{\tau + \frac{1}{n}} = \mathbf{A}_t^\tau + \frac{1}{n} \mathbf{v}_\pi(\mathbf{A}_t^\tau, \mathbf{o}_t, \tau), \tag{1}$$

where $\mathbf{o}_t$ is the current observation. After iterating over $\tau \in [0, 1]$, the final state $\mathbf{A}_t^1$ is used as the predicted action chunk.

*Synchronous inference* is the conventional paradigm adopted in many works (Zhao et al., 2023; Black et al., 2024; Bjorck et al., 2025), where the robot executes all actions within the current execution horizon before supplying the latest observation to the VLA to infer the next chunk (Fig-

ure 1(a)). For synchronous inference to achieve real-time execution, the condition $\delta < \Delta t$ (i.e., $d = 0$) must hold, which is practically unattainable. For example, with a 50 Hz control frequency ($\Delta t = 20$ ms) and $\pi_0$ (Black et al., 2024) as the VLA model, action generation alone requires 76 ms on an NVIDIA RTX 4090 GPU, with additional overhead from preprocessing, disk I/O, and network transmission further increasing latency. As control frequencies rise for fine-grained tasks and larger VLA models are deployed, the effect of inference delay only becomes more pronounced. Consequently, synchronous inference is fundamentally constrained, resulting in jerky transitions between chunks and extended execution times that undermine policy effectiveness.

***Asynchronous inference***, in contrast, achieves real-time execution by ensuring that actions are always available (Figure 1(b)). While this produces smoother trajectories, it introduces two challenges that degrades performance: ❶ **Inter-chunk discontinuity.** Let $\mathbf{A}_t^1$ and $\mathbf{A}_{t+h}^2$ denote two consecutive action chunks sampled from trajectories $\mathcal{T}_1$ and $\mathcal{T}_2$ with execution horizon $h$. If $\mathcal{T}_1$ and $\mathcal{T}_2$ coincide up to timestep $t$ but diverge thereafter, inter-chunk discontinuity arises at the boundary, yielding incoherent transitions between chunks. Intuitively, because an action chunk represents only a local segment of a trajectory, $\mathbf{A}_t^1$ and $\mathbf{A}_{t+h}^2$ may originate from different latent expert modes. Although both $\mathcal{T}_1$ and $\mathcal{T}_2$ are valid continuations of the same past, switching between these modes at the chunk boundary can introduce a large jump in the action sequence, producing jerky or out-of-distribution motion during execution. ❷ **Intra-chunk inconsistency.** Assume the policy $\mathbf{v}_\pi(\mathbf{A}_t|\mathbf{o}_t)$ perfectly captures the underlying environment dynamics and therefore yields the optimal action sequence. Given perception $\mathbf{o}_t$, the optimal executed chunk should fully correspond to $\mathbf{A}_t$. However, under inference delay $d$ with execution horizon $h$, the first $d$ actions executed are instead taken from $\mathbf{A}_{t-h}$. This results in intra-chunk inconsistency, where these inherited prefix actions become suboptimal for the current state, because they were conditioned on $\mathbf{o}_{t-h}$ rather than $\mathbf{o}_t$, creating a perception–action mismatch within the chunk. Figure 1(c) illustrates an example with $P = 3$, $h = 2$, and $d = 1$, highlighting both challenges. Note that inter-chunk discontinuity also arises under synchronous inference, but is further exacerbated in asynchronous settings.

## 4 METHODOLOGY

We consider a **flow-matching policy** $\mathbf{v}_\pi(\mathbf{A}_t|\mathbf{o}_t)$. Our objective is to mitigate both intra-chunk inconsistency and inter-chunk discontinuity by learning a delay-aware policy $\hat{\mathbf{v}}_\pi(\mathbf{A}_t|\mathbf{o}_t, d)$, built upon $\mathbf{v}_\pi(\mathbf{A}_t|\mathbf{o}_t)$. By conditioning on the inference delay $d$, the new policy learns to predict action chunks that are reliable despite the uncertainty introduced by delayed execution. The following sections describe the components of our approach in detail.

### 4.1 MASKED ACTION CHUNKING

Given the pretrained policy $\mathbf{v}_\pi(\mathbf{A}_t|\mathbf{o}_t)$, asynchronous inference executes the first few actions from the previous chunk $\mathbf{A}_{t-h}$, while the remaining actions are drawn from $\mathbf{A}_t$, where $h$ denotes the execution horizon. This misalignment between perception and execution induces a train–test mismatch: supervision on the unexecuted early actions during pretraining can provide misleading signals, leading to off-distribution priors. To address this issue, we introduce REMAC, which incorporates the following learning strategies. The details are described below and summarized in Alg. 1.

**Prefix Masking.** We begin by shifting the learning emphasis toward the *to-be-executed* action chunk segment, only focusing on the portion of the trajectory that directly influences execution while leaving the unexecuted prior intact. Formally, for the predicted flow $\hat{\mathbf{u}} \in \mathbb{R}^{P \times D}$ from $\hat{\mathbf{v}}_\pi(\mathbf{A}_t|\mathbf{o}_t, d)$ and the ground-truth target $\mathbf{u} \in \mathbb{R}^{P \times D}$, we introduce a delay-conditioned *prefix mask* $\mathbf{m}_d$ that restricts supervision to the executable portion of each chunk:

$$\mathbf{m}_d = \{m_d^\tau\}_{\tau=0}^{P-1} = \mathbf{1}[\tau \geq d], \tag{2}$$

where $\tau$ indexes the timesteps within the chunk, $\mathbf{1}[\cdot]$ is the indicator function, and $d \sim \mathcal{U}\{0, \ldots, P-1\}$ is a uniformly sampled random inference delay. By incorporating the prefix mask into the conventional flow-matching loss, we obtain the following objective:

$$\mathcal{L}_\mathrm{m} = \sum_d \frac{\sum_{\tau=0}^{P-1} m_d^\tau ||\hat{\mathbf{u}}_\tau - \mathbf{u}_\tau||_2^2}{\max\left(1, \sum_{\tau=0}^{P-1} m_d^\tau\right)}. \tag{3}$$

This formulation directly builds on the masking strategy, restricting supervision to the unmasked segment while excluding the already-committed prefix. By randomly sampling across all valid inference delays, the policy is exposed to the full spectrum of conditions—from the trivial unmasked case ($d = 0$) to extreme masking scenarios ($d = h$). This mitigates intra-chunk inconsistency by strengthening the policy's adaptability to uncertainty in unexecuted actions, yielding rollouts that remain resilient even under imperfectly predictable environment dynamics. Moreover, the strategy enhances the policy's predictive capacity and equips a single model to handle all valid delay settings, removing the need to train separate policies for different delays.

**Self-conditioned Curriculum.** The actions executed within the inference delay during rollout can serve as informative test-time priors for refining subsequent action chunks. However, during training the executed portion of the chunk is unavailable, unless the current policy is rolled out from scratch for each input sample—a strategy that is prohibitively expensive and impractical, even in simulation. To address this limitation, we introduce a self-conditioned curriculum scheduling method that leverages the pretrained policy to imitate test-time conditions, thereby improving both intra-chunk consistency and inter-chunk discontinuity.

As a concrete example, consider standard flow matching, where the training input is constructed by linearly interpolating between Gaussian noise and the ground-truth action chunk $\mathbf{A}_t$. Instead of directly using the ground-truth chunk, we modify the formulation to incorporate self-conditioning from the pretrained policy, defined as:

$$\hat{\mathbf{A}}_t = \gamma \mathbf{A}_t + \mathrm{sg}((1 - \gamma)\tilde{\mathbf{A}}_t), \qquad \gamma \sim \mathrm{Bernoulli}(\sigma), \sigma \in [0, 1]. \tag{4}$$

Here, $\tilde{\mathbf{A}}_t$ denotes the action chunk predicted by the pretrained policy, $\mathrm{sg}(\cdot)$ is the stop-gradient operation, avoiding unnecessary gradient updates. The mixing parameter $\sigma$ is scheduled based on the training progress, annealing from $\sigma = 1$ (pure ground-truth input) to $\sigma = 0$ (pure self-conditioned input). The resulting mixture $\hat{\mathbf{A}}_t$ is then used in place of $\mathbf{A}_t$ for interpolation.

Therefore, the model input evolves during training: in the early stages, ground-truth actions serve as stable anchors, while in later stages the model is gradually conditioned on its own predictions yet still required to produce accurate outputs. This scheduled curriculum not only stabilizes training, but also mitigates exposure bias by aligning training inputs with test-time conditions. In doing so, it enables the model to learn corrective adjustments over the pretrained policy, refining its own proposals and improving robustness to distribution shifts and compounding rollout errors. Figure 7 illustrates several scheduling functions for $\sigma$, among which we adopt the piecewise linear decay as the default.

---

**Algorithm 1** REMAC

**Input:** Delay set $S_d$, dataloader $D$, epochs $E$
**Initialize:** Pretrained policy $\mathbf{v}_\pi(\cdot)$, target policy $\hat{\mathbf{v}}_\pi(\cdot)$

1: **for** $e \leftarrow 0$ **to** $E - 1$ **do**
2:     Sample $d \in S_d$ and compute $\mathbf{m}_d$ via Eq. 2.
3:     **for all** $(\mathbf{o}_t, \mathbf{u}_t) \in D$ **do**
4:         $\tilde{\mathbf{A}}_t \leftarrow \mathrm{Integrate}(\mathbf{v}_\pi(\mathbf{o}_t))$
5:         $\hat{\mathbf{A}}_t \leftarrow \mathrm{Interpolate}(\tilde{\mathbf{A}}_t, \mathbf{u}_t)$     ▷ Eq. 4
6:         $\hat{\mathbf{u}}_t \leftarrow \hat{\mathbf{v}}_\pi(\mathbf{o}_t, \hat{\mathbf{A}}_t)$     ▷ LoRA enabled
7:         $\tilde{\mathbf{u}}_t \leftarrow \mathbf{v}_\pi(\mathbf{o}_t, \hat{\mathbf{A}}_t)$     ▷ LoRA disabled
8:         Update $\hat{\mathbf{v}}_\pi$ by minimizing Eq. 6
9:     **end for**
10: **end for**

---

**Residual Alignment.** In addition to standard supervision against the ground truth, we introduce a $\Delta$-*matching* term on the unmasked action chunk. This term explicitly aligns the induced correction with the residual between the pretrained policy's prediction and the ground-truth target. Specifically, let $\tilde{\mathbf{u}}$ denote the flow estimate from the pretrained backbone $\mathbf{v}_\pi(\mathbf{A}_t|\mathbf{o}_t)$. The correction is then encouraged to match the residual toward the target through the following objective:

$$\mathcal{L}_\Delta = \sum_d \frac{\sum_{\tau=0}^{P-1} ||m_d^\tau(\mathbf{u}_\tau - \tilde{\mathbf{u}}_\tau) - m_d^\tau(\hat{\mathbf{u}}_\tau - \tilde{\mathbf{u}}_\tau)||_2^2}{\max\left(1, \sum_{\tau=0}^{P-1} m_d^\tau\right)}. \tag{5}$$

While mathematically related to Eq. 3, the two objectives emphasize different aspects: Eq. 3 enforces direct alignment with the ground truth, whereas Eq. 5 explicitly models the residual adjustment relative to the pretrained policy. Empirically, we observe that incorporating $\mathcal{L}_\Delta$ into training yields substantial performance improvements.

Finally, the overall training objective is defined as:

$$\mathcal{L} \;=\; \lambda_{\mathrm{m}} \mathcal{L}_{\mathrm{m}} + \lambda_{\Delta} \mathcal{L}_{\Delta}, \tag{6}$$

where $\lambda_{\mathrm{m}} = 0.01$ and $\lambda_{\Delta} = 0.01$ are the weighting coefficients.

## 4.2 PREFIX-PRESERVED SAMPLING

Having obtained the new policy $\hat{\mathbf{v}}_{\pi}(\mathbf{A}_t|\mathbf{o}_t, d)$, we further adjust the sampling pipeline to align with the training procedure and enhance inter-chunk continuity. Firstly, the initial action state $\mathbf{A}_t^0$ is no longer drawn from the Gaussian prior. Instead, it is initialized using the can be executed actions during the delayed time steps, denoted as $\mathbf{A}_t^{\mathrm{P}} \in \mathbb{R}^{P \times D}$, whose first $P - h$ entries are filled with the last $P - h$ actions from the previous predicted chunk, while the remaining entries are zero-initialized.

Secondly, during the sampling process, the overlapping segment between the currently executing chunk and the upcoming chunk are preserved as the executed ones, and only the remaining portion of the action chunk is newly synthesized. Specifically, if the sampling process integrates the learned velocity field $\hat{\mathbf{v}}_{\pi}$ over $n$ integration steps, let $\mathbf{m} \in \{0, 1\}^P$ denote the delay-conditioned prefix mask and $\mathbf{A}_t^{\tau}$ is the intermediate action state at integration time $\tau$, then:

$$\mathbf{A}_t^{\tau + \frac{1}{n}} \;=\; \mathbf{m} \odot \left(\mathbf{A}_t^{\tau} + \tfrac{1}{n}\hat{\mathbf{v}}_{\pi}(\mathbf{A}_t^{\tau}, \mathbf{o}_t, \tau)\right) \;+\; (1 - \mathbf{m}) \odot \mathbf{A}_t^{\mathrm{P}}, \tag{7}$$

where $\odot$ denotes element-wise multiplication. A special case arises for the first action chunk generation at rollout initialization, where no previous actions have yet been executed. In this case, we set $\mathbf{A}_t^{\mathrm{P}} = \mathbf{0}$ or random Gaussian noise to indicate the absence of prior actions, and apply the standard integration rule in Eq. 1.

## 4.3 IMPLEMENTATION DETAILS

**Model Adaptation.** We adapt the pretrained policy using the parameter-efficient finetuning technique LoRA (Hu et al., 2021), introducing at most 1.5% additional parameters relative to the original model. The reasons we adopt LoRA instead of full model finetuning are twofolds. First, we conceptualize the transformation from $\mathbf{v}_{\pi}(\mathbf{A}_t|\mathbf{o}_t)$ to $\hat{\mathbf{v}}_{\pi}(\mathbf{A}_t|\mathbf{o}_t, d)$ as a distributional adjustment with respect to the pretrained policy, rather than a wholesale re-learning of the policy under a different objective. Second, LoRA offers a flexible mechanism that can leave the original parameters untouched, thereby preserving the predictive capacity of the pretrained model and making it well suited to our method. During implementation, the LoRA module can be further extended with an additional projection matrix that incorporates the prefix mask to enrich the input representation.

**Training and Sampling.** During training, we control the prefix-mask coverage by uniformly sampling $d$ in Eq. 2 from a gradually shrinking interval $[q, q_{\max}]$, where $q_{\max}$ is fixed and $q$ is linearly annealed from $q_{\max}$ down to a final $q_{\min}$. Importantly, $q_{\max}$ and $q_{\min}$ are *not* limits on the inference delays that the model can handle; rather, they are hyper-parameters that determine the strength of mask-induced perturbations during training. The total loss defined in Eq. 6 is then used to update the LoRA parameters. During action sampling, compared to the conventional approaches in prior works (Chi et al., 2024; Black et al., 2024), the model additionally takes the pre-determined or estimated inference delay and the previous action chunk as inputs for the next action chunk prediction. The LoRA modules can be further merged into the backbone model layers, such that no additional time cost is introduced.

**Deployment Framework.** For simulation benchmarks, we follow the setting of Black et al. (2025), in which the predicted actions are deliberately segmented and concatenated according to predefined delays and execution horizons. For the real-world deployment, we adopt a setup similar to Shukor et al. (2025), in which actions are predicted on a remote server and transmitted to the robot via gRPC. The robot maintains a queue of actions to execute while sending observations to the server at the same frequency as its control loop. The remote server, hosting the VLA model, likewise maintains a queue of incoming observations and performs action prediction as frequently as possible using the most recent inputs and states. The inference delay is estimated on the robot client side as the maximum of the most recent few measured delays.

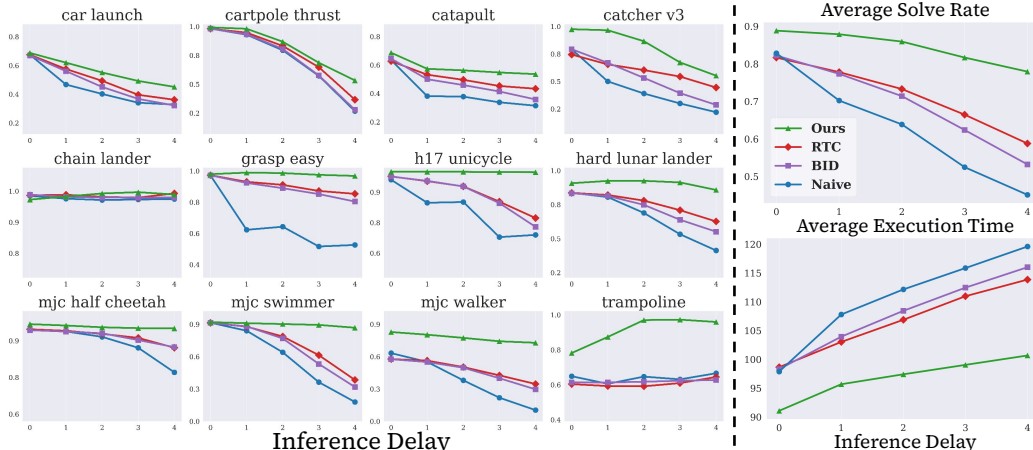

Figure 2: Performance comparison in Kinetix environments. **Left:** Solve rates for individual tasks under varying inference delays. **Right (top):** Average performance across all environments. Our method consistently outperforms baselines under all delay settings and exhibits smaller performance degradation as delay increases. **Right (bottom):** Average execution time across all environments. Our method requires fewer steps and achieves faster task completion.

## 5 EXPERIMENTS

In this section, we present experiments evaluating the effectiveness of our method against baseline approaches under various inference delay settings. We also perform ablation studies on different components to assess their individual contributions. Our empirical analysis spans both simulation and real-world tasks, providing a comprehensive evaluation of the proposed method.

### 5.1 KINETIX ENVIRONMENT

We first evaluate our method on the Kinetix simulator (Matthews et al., 2025), following the protocol of Black et al. (2025). The benchmark comprises 12 highly dynamic and stochastic environments specifically designed to test asynchronous execution. Expert policies are first trained under a binary success reward using RPO (Rahman & Xue, 2022), from which training data are collected via demonstration generation. Flow-matching policies are then learned through imitation learning (Zare et al., 2023; Argall et al., 2009). For each task, the trained policies are configured with a prediction horizon of 8. The inference delay $d$ ranges from 0 to 4, and the execution horizon for each delay is chosen within $\max\{1, d\}$ to $9 - d$, ensuring continuous action availability without gaps between consecutive chunks. LoRA is applied to all linear layers except the time-embedding and AdaLN layers (Peebles & Xie, 2023), with each LoRA layer assigned a rank of 4. For each fixed delay value, evaluation metrics are reported as averages over all corresponding execution horizon settings.

**Baselines.** The baselines we compare with include: (1) **Naive Async**, which directly uses the pretrained policy and executes actions from the most recently generated action chunk. (2) **Bidirectional Decoding (BID)** (Liu et al., 2025), a test-time method that samples multiple candidate predictions and applies rejection sampling to select the optimal one, aiming to balance long-term consistency with short-term reactivity. (3) **RTC** (Black et al., 2025), also a test-time execution strategy that leverages an inpainting algorithm (Pokle et al., 2024). It applies gradient-based corrections to the predicted actions, using the executed actions during the inference delay as priors. We omit the baseline of Temporal Ensembling (TE) (Zhao et al., 2023), as both our experiments and Black et al. (2025) show that TE substantially underperforms the other baselines in this simulation benchmark, even falling behind Naive Async.

**Results.** Figure 2 reports the per-task and average performance across 12 tasks, measured by both task success rate and completion time. Generally, we observe a consistent decline in performance as inference delay increases across all methods. This degradation stems from the growing mismatch between actions and observations, which amplifies intra-chunk inconsistency, and from the increasing influence of the previous chunk, which exacerbates inter-chunk discontinuity. Our method consis-

Table 1: Effectiveness of the different components in REMAC.

| Method | $d=0$ | $d=1$ | $d=2$ | $d=3$ | $d=4$ |
|---|---|---|---|---|---|
| Naive | 0.828 | 0.702 | 0.639 | 0.525 | 0.451 |
| + LoRA | 0.825 | 0.710 | 0.630 | 0.510 | 0.428 |
| + Prefix masking | 0.863 | 0.825 | 0.752 | 0.729 | 0.636 |
| + Self-conditioned curriculum | 0.848 | 0.837 | 0.805 | 0.762 | 0.710 |
| + $\mathcal{L}_\Delta$ (**Ours**) | **0.888** | **0.879** | **0.859** | **0.817** | **0.779** |

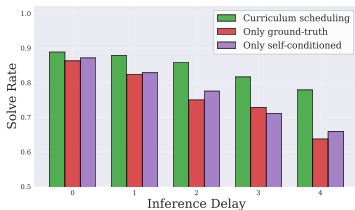

Figure 3: Schedule comparison.

tently outperforms all baselines across all delay settings, with especially pronounced gains under larger inference delays. It also achieves shorter average execution time than competing methods, demonstrating the ability to produce more efficient policies for faster task completion.

Our method also improves performance under $d = 0$. We attribute this to the masked action chunking formulation, which encourages the policy to enforce stronger coherence and temporal dependencies even without inference delay, thereby enhancing its predictive ability. Moreover, as the inference delay increases, our

Table 2: Integration with test-time methods.

| Method | $d=0$ | $d=1$ | $d=2$ | $d=3$ | $d=4$ |
|---|---|---|---|---|---|
| Ours | 0.888 | 0.879 | 0.859 | 0.817 | 0.779 |
| + BID | 0.888 | 0.880 | 0.862 | 0.821 | 0.781 |
| + RTC | 0.888 | 0.879 | 0.864 | 0.826 | 0.791 |

method exhibits both a smaller performance drop and a lower rise in execution time compared to baselines. This trend highlights the robustness of our approach under increasing delays, underscoring its potential as a versatile solution for asynchronous inference.

**Ablations and Analysis.** We conduct ablation studies to evaluate the contribution of each component and further analyze the extensibility of our approach.

- Table 1 reports the contribution of each component in our method. First, adding LoRA alone—without modifying the training paradigm—yields no performance gain, indicating that the effectiveness of our approach cannot be attributed merely to the increase in parameters. In contrast, progressively incorporating the components described in Sec. 4.1 leads to consistent improvements, with the full method achieving the highest overall success rate. Detailed per-task results are provided in Sec. D.

- Figure 3 compares variations of the self-conditioned curriculum schedule introduced in Eq. 4. We compare with two baselines: one that initializes exclusively with ground-truth actions ($\sigma = 1$) and another that relies entirely on self-conditioned inputs ($\sigma = 0$). The results show that the curriculum schedule improves both performance and training stability relative to these extremes. Training with pure ground-truth inputs suffers from exposure bias, since during policy rollout the sampling process relies on the model's own predictions. Conversely, training with fully self-conditioned inputs often destabilizes early learning. The curriculum schedule mitigates both issues by gradually transitioning from ground-truth to self-conditioned inputs.

- Our method can further be integrated with other test-time approaches such as BID and RTC, since it only modifies the backbone policy. Table 2 shows that, although the improvements are modest, integration consistently provides additional performance gains across delay settings, with larger improvements observed under higher delays. This demonstrates both the compatibility of our approach with existing test-time strategies and its potential as a plug-and-play method.

- We also ablate the choice of $q_{max}$ and $q_{min}$ (Sec. 4.3) to assess how mask coverage affects performance. As shown in Fig. 10(a), performance changes only marginally across configurations, indicating that REMAC is not highly sensitive to these hyper-parameters. Larger values—particularly a larger $q_{min}$—produce slightly worse results. We use $q_{max} = 4$ and $q_{min} = 0$ in practice.

- We further show that REMAC is not limited to flow-matching policies. In Sec. E.5, we integrate REMAC into the Transformer-based ACT (Zhao et al., 2023) framework, where it consistently outperforms both the naive asynchronous baseline and the LoRA-only baseline. These results demonstrate that REMAC readily extends beyond flow matching and can be seamlessly incorporated into diverse action–chunking architectures.

## 5.2 REAL-WORLD ENVIRONMENT

**Setup.** We employ a Franka Research 3 robot (7-DoF arm) (Haddadin, 2024) equipped with parallel-jaw grippers and adopt the DROID setup (Khazatsky et al., 2025) (Figure 6). The additional

Table 3: Average completion progress. Progress is measured by discrete scores corresponding to the sub-tasks completed.

| Method | Grasp-Easy | Grasp-Medium | Grasp-Hard |
|---|---|---|---|
| Synchronous | 0.805 | 0.718 | 0.670 |
| Naive (Shukor et al., 2025) | 0.825 | 0.825 | 0.460 |
| TE (Zhao et al., 2023) | 0.825 | 0.868 | 0.717 |
| RTC (Black et al., 2025) | 0.823 | 0.848 | 0.753 |
| **Ours** | **0.903** | **0.943** | **0.812** |

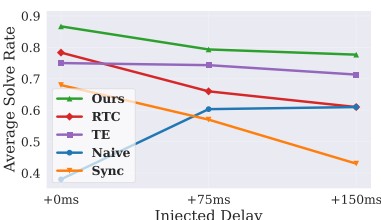

Figure 4: Performance under delay injections on *Grasp-Hard*.

tactile sensors of the grippers are disabled and repurposed to narrow the gripping range, thereby acting as constraints for evaluating fine-grained control. This design ensures that the tasks demand precise action execution, making them well suited for assessing performance under asynchronous inference. For the backbone VLA, we use $\pi_0$ (Black et al., 2024), configured with a prediction horizon of $P = 50$, and adopt its memory-efficient variant for finetuning. A total of 200 trajectories are collected for model adaptation, with LoRA layers of rank 8 inserted only into the action expert module. After finetuning, the LoRA weights are merged into the backbone, ensuring that no additional computational overhead is introduced at inference.

During execution, the robot operates at a control frequency of 15Hz, corresponding to $\Delta t \approx 67$ms. The execution horizon is fixed at $h = 8$, and sampling is performed over 10 steps. Processing through the VLA model **without** any test-time strategies takes approximately 76-80ms per observation. Since the model is hosted on a separate server, communication over LAN introduces an additional network delay of 34–40ms, while data processing and disk writing contribute a further 10–20ms. In total, the end-to-end inference delay is roughly 122–140ms, corresponding to an effective inference delay of $d = 2$ or 3.

**Task Design and Measurement.** We evaluate our approach on three single-arm grasp-and-place tasks (*Grasp-Easy, Medium, Hard*) of varying difficulty. The tasks range from manipulating simple objects, such as a cucumber, to more challenging ones, such as a Rubik's cube whose sides are only 1cm shorter than the gripper's jaw gap, requiring precise control. Placement targets are either a plate or a bowl, with the latter posing greater difficulty and demanding finer manipulation accuracy.

Task performance is measured using a stage-based solve rate that evaluates progress through four steps: (1) reaching the object, (2) gripping and lifting it, (3) moving it toward the target location, and (4) placing it correctly into the container. For each task, we conduct 30 evaluation trials per method, with each trial capped at 300 steps, amounting to a total of 6 hours of robot execution time. The initial position and orientation of the objects are randomized across trials.

**Baselines.** We compare our method against four baselines: (1) **Synchronous inference**, the widely used baseline in prior works (Black et al., 2024; Kim et al., 2024; Hu et al., 2025; Kim et al., 2025; Pertsch et al., 2025), which executes an entire predicted action chunk and then pauses until new actions are received. (2) **Naive Async**, and sampling is run as frequently as possible such that the most recent actions are queued. (3) **Temporal Ensembling** (Zhao et al., 2023), an extension of Naive Async that aggregates overlapping actions across consecutive chunks by weighted averaging. (4) **RTC**, the state-of-the-art baseline but it introduces an additional $55 - 64$ms of inference latency. We omit BID from comparison, as it is substantially more time- and computation-intensive than the other methods (Black et al., 2025), and is therefore not a competitive baseline in real-world settings.

**Results.** Table 3 reports the average completion progress for each task, showing that our method achieves higher completion rates across all tasks. During execution, synchronous inference produces frequent pauses, often leading to unintended object drops and inaccurate localization. Asynchronous baselines generate smoother trajectories without pronounced jerkiness. However, Naive Async and Temporal Ensembling remain prone to premature or delayed grasping and placement. In contrast, RTC suffers from the additional inference delay it introduces.

Figure 4 reports results with additional latency injections of 75ms and 150ms, simulating deployment under slower hardware and network conditions. Even with total inference delays of $3 - 5$, our method consistently outperforms all baselines, demonstrating robustness to varying delay levels. Interestingly, Naive Async performs comparatively better under larger delays, while RTC exhibits

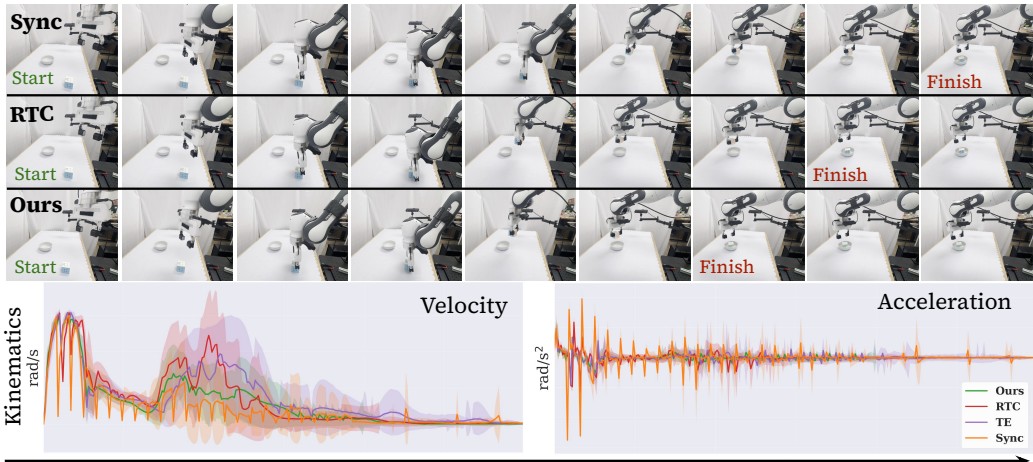

Figure 5: Visual comparison of different methods under a 150ms injected delay. **Top**: Task completion progress, where our method achieves faster completion. **Bottom**: Average robot kinematics during task execution on *Grasp-Hard*. Our method exhibits smoother trajectories and faster task completion speed.

significantly degraded performance. We attribute this to the fact that larger delays correspond to longer execution horizons in real-world settings: less frequent chunk switching reduces inter-chunk discontinuity, but the test-time adjustments made by RTC can have adverse effects.

Figure 5 compares task completion progress and corresponding robot kinematics. We evaluate under an injected 150ms delay and adopt cases where all policies achieve successful rollouts, ensuring fair comparison while amplifying differences across methods. Qualitatively, our method completes tasks within a shorter time. Quantitatively, analysis of average robot velocity and acceleration over 15 trials shows that synchronous inference produces abrupt, periodic kinematic changes, whereas asynchronous inference methods yield smoother trajectories. Among these, our method achieves the most stable dynamics with fewer abrupt changes, highlighting both speed and stability.

## 6 CONCLUSION

In this paper, we address the problem of effective real-time robot manipulation under asynchronous inference. We identify two critical challenges in this setting—exacerbated inter-chunk discontinuity and intra-chunk inconsistency—and propose REMAC to mitigate them. Unlike prior test-time approaches, REMAC learns corrective adjustments on top of a pretrained policy through a masked action chunking strategy and a prefix-preserved sampling pipeline, while introducing no additional inference delay. Extensive experiments in both simulation and real-world benchmarks demonstrate that our method is robust to varying delay conditions and achieves faster task completion.

## ETHICS STATEMENT

This work focuses on algorithmic development and evaluation for asynchronous inference in robotic control. All experiments were conducted either in simulation or with a physical Franka Research 3 robotic arm in a controlled laboratory setting. Our method is intended solely for research purposes and does not present foreseeable risks of harmful deployment. We believe our work fully adheres to the ethical standards and guidelines of the community.

## REPRODUCIBILITY STATEMENT

To ensure reproducibility, we provide a detailed algorithmic description in Alg. 1 and implementation choices in Sec. 4.3. In the experimental section, we clearly specify the datasets, training protocols, evaluation metrics, and implementation details, including inference settings. In addition, source code is released.

ACKNOWLEDGEMENT

This research is supported by NSF IIS-2525840, CNS-2432534, ECCS-2514574, NIH 1RF1MH133764-01 and Cisco Research unrestricted gift. This article solely reflects opinions and conclusions of authors and not funding agencies. We gratefully acknowledge the individuals who provided valuable guidance and feedback for this project, including Changda Tian (FORTH) for hardware-related suggestions and Hongkai Zheng (Caltech) for insightful advice on inpainting.

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

## A    HARDWARE SETUP

Figure 6 illustrates our robot setup, where the gap between the gripper jaws is deliberately reduced to increase task difficulty and require finer control. Although two third-view cameras are mounted, only one third-view camera and the wrist-mounted camera are used for data processing. This configuration provides both a global view of the scene and a local, fine-grained perspective of the manipulation area, ensuring accurate observation while maintaining a controlled evaluation environment.

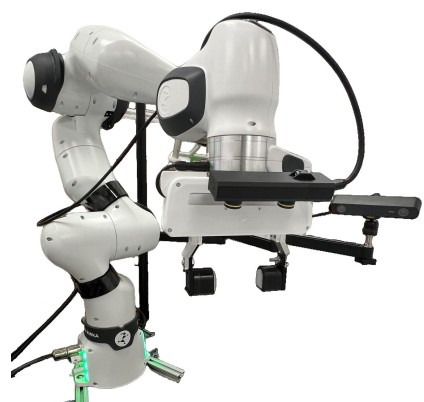

Figure 6: Robot setup illustration.

## B    SCHEDULING FUNCTION VARIANTS

Figure 7 illustrates different variants of the scheduling function $\sigma$. While multiple options are available, we primarily adopt the piecewise linear schedule, as it provides a smoother warm-up phase and more stable optimization compared to other choices.

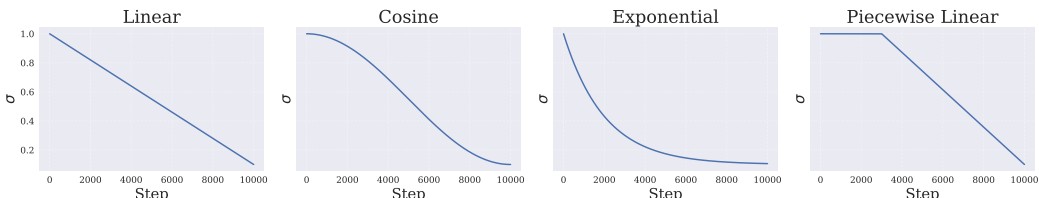

Figure 7: Different scheduling functions.

## C    TASK EXAMPLES

Figure 8 presents wrist-camera views of the tasks included in our experiments. These examples highlight the visual perspectives used for policy input and illustrate the varying levels of manipulation difficulty across tasks.

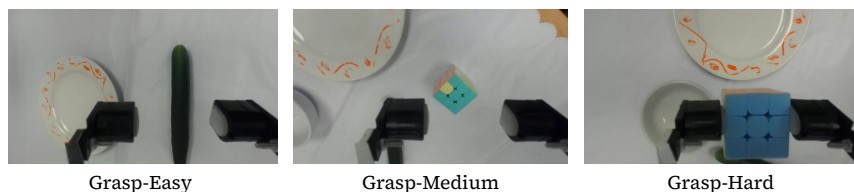

Figure 8: Examples of our included tasks.

## D    PER-TASK SIMULATION RESULTS

Figure 9 presents the per-task results corresponding to the ablations in Table 1. The detailed metrics demonstrate that each component of our method contributes consistent improvements in success rate across tasks, highlighting the generalizability of the design choices.

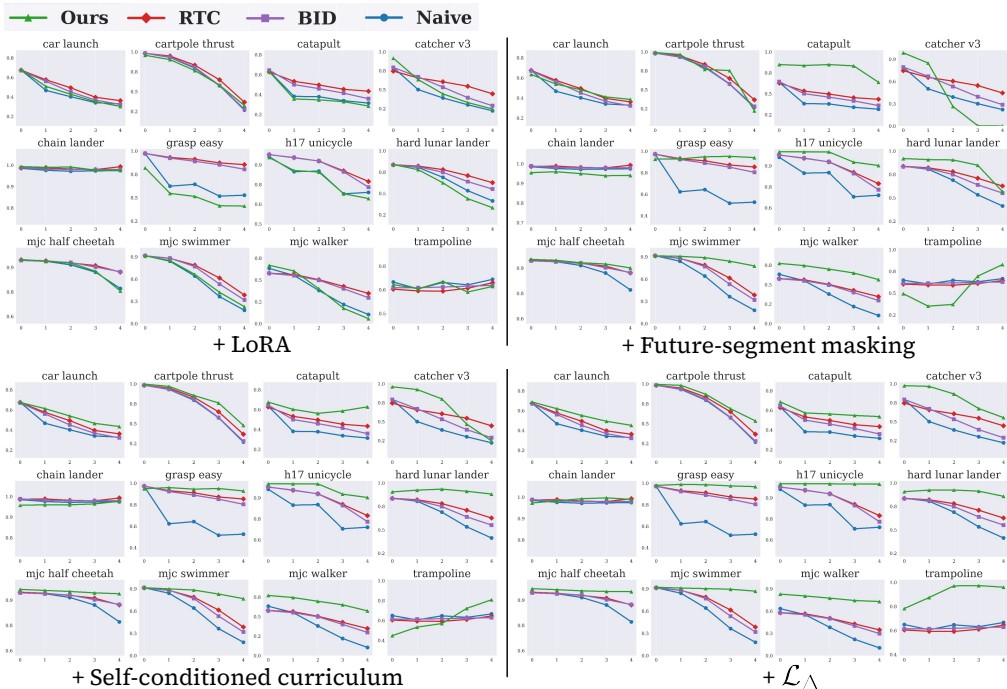

Figure 9: Per-task results for component ablations.

## E    ADDITIONAL ABLATIONS

### E.1    EFFECT OF $q_{MAX}$ AND $q_{MIN}$

In Figure. 10(a), we evaluate the influence of the hyperparameters $q_{max}$ and $q_{min}$ (Sec. 4.3) by testing multiple combinations under the same training setup. Overall, we observe only minor performance variation across different settings, indicating that REMAC is not sensitive to the exact choice of these values. However, larger values - particularly a larger $q_{min}$ - tend to produce slightly worse performance. This is expected: when $q_{min} > 0$, the prefix of length $q_{min}$ is always masked during training, which encourages the model to behave over-conservatively during rollout. In practice, we adopt $q_{max} = 4$ and $q_{min} = 0$. Although this choice happens to coincide with the delay range evaluated in our simulation experiments, it should not be interpreted as limiting the range of delays REMAC can handle.

### E.2    EFFECT OF LEARNING A MASK EMBEDDING

Our method conditions on the inference delay by converting it into a prefix mask that is applied during both training and inference. Beyond its role in loss computation and sampling, this mask can also be treated as an additional input signal by projecting it into a learnable mask embedding and injecting it into the model. In Figure 10(b), we evaluate the effect of introducing such a learned mask embedding under the same training setup. The results show mixed but generally stable outcomes: while a few tasks see marginal changes, most tasks retain similar performance. This indicates that REMAC is robust to architectural variations and does not depend critically on whether the delay mask is embedded or used directly.

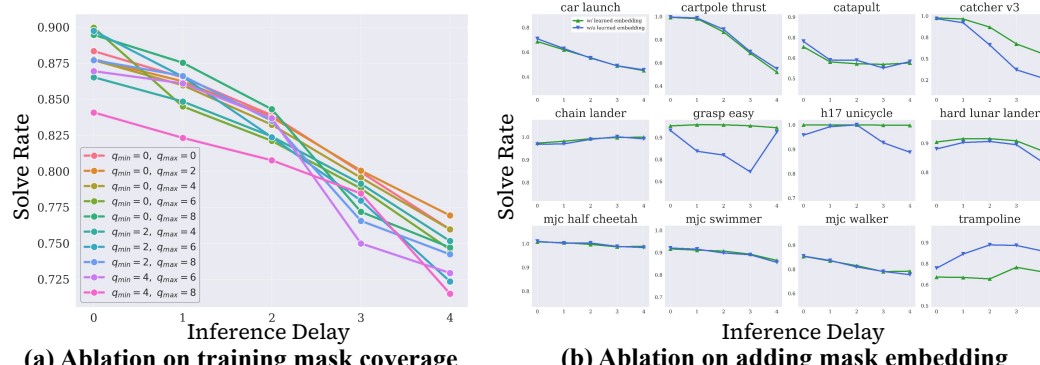

**(a) Ablation on training mask coverage**   **(b) Ablation on adding mask embedding**

Figure 10: **(a)** Ablation on the effect of different $q_{max}$ and $q_{min}$. **(b)** Ablation on the effect of adding mask embeddings as additional information.

## E.3 DATA-EFFICIENCY OF REMAC

To evaluate sensitivity to dataset size, we conducted additional real-world experiments using only 10 demonstrations per task, compared to the total 200 demonstrations used in the main paper. The average completion progress (10 evaluations per task) is shown in Table 4 below:

Table 4: Effect of Training Data Quantity

| #Traj | Grasp-Easy | Grasp-Medium | Grasp-Hard |
|---|---|---|---|
| 200 | 0.910 | 0.936 | 0.820 |
| 30 | 0.900 | 0.905 | 0.810 |

Even with much less data, REMAC maintains performance very close to the 200-trajectory model, with only mild degradation. These results indicate that REMAC is not data-hungry and remains effective in low-data regimes, thanks to its self-supervised masking mechanism that exposes the model to diverse prefix deviations without requiring additional demonstrations.

## E.4 EFFECT OF REMAC ON GENERALIZABILITY

We further examine whether REMAC affects the policy's generalization ability. Our fine-tuning pipeline is as follows: we first fine-tune $\pi_0$ using our collected demonstrations following the official implementation, updating both the VLM backbone and the action expert. This fine-tuned $\pi_0$ serves as the baseline model for all methods (Naive, RTC, BID, and REMAC). REMAC then applies LoRA *only* to the baseline model's action expert, while keeping the entire VLM backbone frozen.

We evaluate generalizability under two settings: (i) scene/background variation and (ii) novel language prompts. For background variation, we remove curtains, replace the table covering, and introduce additional distractor objects. REMAC behaves similarly to the baseline model: both policies reliably ground objects seen during fine-tuning despite the visual changes and background noise. However, under unseen language instructions, both the baseline and REMAC fail to perform novel tasks or recognize unseen objects (e.g., "cup," "box"), suggesting that the limitation stems from the underlying fine-tuned policy rather than from REMAC.

We additionally evaluate on $\pi_{0.5}$ (Intelligence et al., 2025b), which exhibits stronger open-world generalization under our settings. Under unseen language prompts, the fine-tuned $\pi_{0.5}$ correctly recognizes unseen objects but still struggles with unseen tasks. Applying REMAC on top of $\pi_{0.5}$ preserves this OOD capability and does not introduce noticeable degradation, confirming that REMAC does not harm the generalization already present in the underlying VLA model.

In summary, REMAC performs only low-rank adjustments on the action expert and leaves the grounding and perception capabilities of the VLM untouched. As a result, it does not diminish the generalizability of the fine-tuned base model.

### E.5 APPLICATION TO OTHER POLICY CLASSES

We further demonstrate that REMAC is *not* restricted to flow-matching policies and can be applied to other policy architectures. While diffusion-style models are a natural extension due to their structural similarity to flow matching, we additionally show that REMAC is compatible with **Transformer-based chunking policies**.

To validate this, we integrate REMAC into ACT (Zhao et al., 2023) by applying LoRA to its decoder layers and action head, and replacing the $\ell_2$ losses used in Eq. 3 and 5 with ACT's $\ell_1$ and KL objectives. In Table 5 and 6, we evaluate on two bimanual ACT tasks and report *success rate / average return* under varying delays. Across all delay settings, REMAC consistently outperforms both the naive asynchronous baseline and the LoRA-only baseline. These results indicate that REMAC generalizes beyond flow-matching models and can be seamlessly incorporated into diverse action–chunking frameworks.

Table 5: Transfer Cube (h = 12)

| d | Naive | +LoRA | **Ours** |
|---|---|---|---|
| 4 | 0.40 / 354.44 | 0.52 / 335.04 | **0.74 / 486.78** |
| 6 | 0.48 / 348.76 | 0.58 / 352.24 | **0.72 / 508.28** |
| 8 | 0.46 / 309.32 | 0.68 / 422.62 | **0.68 / 460.86** |

Table 6: Insert Box (h = 30)

| d | Naive | +LoRA | **Ours** |
|---|---|---|---|
| 0 | 0.14 / 230.74 | 0.20 / 218.74 | **0.18 / 245.72** |
| 5 | 0.14 / 219.78 | 0.12 / 179.90 | **0.18 / 217.12** |
| 10 | 0.14 / 216.98 | 0.10 / 183.94 | **0.16 / 220.68** |

## F ROBUSTNESS TO VARYING DELAY CONDITIONS

Real-world latency can be fluctuating and noisy. We conducted additional evaluations in both simulation and the real world to assess REMAC's robustness under noisy, rapidly fluctuating, and adversarially spiky delay patterns.

**(1) Simulation experiments.** We fix the true execution delay to $d$, but deliberately pass incorrect delay values to the policy using the following schemes:

- **Noisy and rapidly fluctuating delays:** For each policy call, we sample the delay from $\{d - 1, d, d + 1\}$ (i.e., 66% inaccurate).
- **Spiky delays:** With 10% probability, we replace the delay with the maximum valid value to simulate infrequent but large latency spikes.

Below we report the average performance over 12 Kinetix tasks:

Table 7: Performance Comparison under Different Latency Conditions.

| Setting | $d{=}0$ | $d{=}1$ | $d{=}2$ | $d{=}3$ | $d{=}4$ |
|---|---|---|---|---|---|
| RTC | 0.817 | 0.778 | 0.733 | 0.665 | 0.588 |
| Ours | **0.877** | **0.860** | **0.832** | **0.796** | **0.760** |
| RTC + Noisy & Fluct. | 0.816 | 0.770 | 0.728 | 0.653 | 0.573 |
| Ours + Noisy & Fluct. | **0.866** | **0.837** | **0.799** | **0.746** | **0.757** |
| RTC + Noisy & Fluct. & Spiky | 0.814 | 0.772 | 0.728 | 0.653 | 0.575 |
| Ours + Noisy & Fluct. & Spiky | **0.820** | **0.796** | **0.769** | **0.718** | **0.757** |

Even under extremely ill-conditioned latency sequences, both RTC and REMAC degrade gracefully—never catastrophically. Importantly, REMAC under corrupted delays still outperforms RTC under accurate delays, demonstrating strong robustness to delay misestimation.

**(2) Real-world experiments.** All of our real-world evaluations already operate under realistic network- and compute-induced delay profiles, where delay measurements are noisy and temporally correlated. The measured delay includes multiple sources—VLA inference time, network transmission, file I/O, memory contention, and system-level scheduling jitter. Because the delay measurement always includes inference time, it is necessarily historical. Following RTC (Black et al., 2025), we use the maximum delay observed in a recent window as the estimate passed to the policy. Thus, all real-world results (Table 3 and Figure 4) inherently reflect noisy and imprecise delay estimates, yet REMAC remains robust and performs strongly under these conditions.

To further stress-test the system, we artificially corrupt the delay input by sampling from $\{d-1, d, d+1, d+2\}$ for every inference step. This induces noisy, fluctuating, and spiky delays. The system exhibits only 1–2 additional failures out of 10 trials, primarily due to occasional overestimation $(d+2)$, which leads the robot to temporarily pause and exceed the 300-step time limit. This is a timeout artifact rather than an indication of policy instability.

**(3) Inherent robustness from discretized delays.** Asynchronous execution exhibits a natural robustness property: the inference delay is defined as

$$d = \left\lfloor \frac{\delta}{\Delta t} \right\rfloor,$$

where $\delta$ is the continuous inference latency and $\Delta t$ is the controller sampling period (67 ms at 15 Hz). For $d$ to fluctuate by $\pm 1$, the underlying latency must shift by more than 67 ms—an extreme fluctuation rarely observed in practice. This discretization smooths noise in $\delta$, making asynchronous methods, including ours, inherently robust to moderate latency variations.

## G   LIMITATIONS

Our method is not without limitations. First, it requires specifying a maximum inference delay in advance to ensure that the optimization process covers the full range of possible delays. If the actual delay during execution exceeds this bound, unexpected failure may occur. Second, the approach may demand a substantial amount of finetuning data for masked finetuning, which could limit practicality in settings where data collection is costly or constrained.

## H   LLM USAGE

We used LLM (ChatGPT) to assist with writing refinement. Specifically, it was employed to improve clarity, grammar, and flow of text, as well as to adjust tone for academic writing. No content generation, experimental design, or analysis was delegated to the LLM; all technical contributions, mathematical definitions, and experimental results were developed by the authors. The LLM's role was limited to language polishing and presentation, and all outputs were carefully reviewed and edited by the authors.

