# OpenReview forum: "Real-Time Robot Execution with Masked Action Chunking"
_ICLR.cc/2026/Conference — ICLR 2026 Poster_

### Official Review · Reviewer_LpbR · 2025-10-26

**Soundness:** 3
**Presentation:** 1
**Contribution:** 3
**Rating:** 6
**Confidence:** 4

**Summary:**

This paper proposed a method for finetuning a pretrained VLA policy toward residual modifications i.e., adjusting an action chunk from the pretrained policy by taking into account delay due to latency. To do so, the paper introduces a curriculum learning method in which the policy first learns to imitate the expert and then learns to do residual corrections conditioned on a wide range of delay values. For sampling at test-time, the model conditions on the actions being executed from the previous chunk (due to delay) and the sampling procedure enables the model to adjust the trajectory towards more optimal actions.

**Strengths:**

1. The proposed method has been tested on a wide range of tasks in both simulation and in the real world. The paper offers interesting analysis at a low-level such as the kinematics of the robot under various methods (see figure 5).
2. The paper compares the proposed method with all the appropriate baselines.
3. The paper carries out several important ablations and offers interesting insights into how baseline methods can further uplift the performance of REMAC.

**Weaknesses:**

1. Lack of clarity in the problem formulation, the definition of inter-chunk discontinuity is very unclear (line 153). It is not clear why two trajectories that diverge must not have actions that are different i.e. if the trajectories are similar only up to time $t$ and then the two trajectories diverge to different observations, their action chunks at time $t + h$ should be different? Furthermore, it is not clear what this discontinuity means and what the expected behavior is if there is no discontinuity. Similarly, intra-chunk inconsistency is unclear. While the first $d$ actions are taken from the previous action chunk $\textnormal{A}_{t-h}$, I do not understand what the perception-action mismatch is.
2. Lack of clarity in the method discussion. Generally, since there are various choices of implementing flow matching, it would be helpful to either have a preliminaries or an appendix section where you define both the predicted flow and the ground truth flow and, in your notation, specify which policy each is being sampled from. I understand that abstracting these modeling choices helps put the emphasis more on the generality of your method but, at least, some concrete examples will help the reader develop a more concrete understanding of your method. Similarly, the switch to using $\textnormal{x}_\mathrm{p}$ to denote the action seems strange. I also think section 4.2 is extremely dense and unclear, from notation to definitions (for e.g., the definition of $f$ in equation 6).
3. While the method discusses handling temporal consistency, it is unclear whether this method handles the latency issues as discussed in the RTC paper (Black et. al. 2025). It seems like the delay-aware policy would also have high latency, so is it that your method relies on the conditioning on the delay to take care of that?

As is clear, most of my complaints are with respect to the clarity of the discussion of the problem formulation and the method. I would be happy to raise my score if these issues with clarity are adequately answered. Apart from them, I have some questions about the scalability of the method.

4. The method seems quite data expensive in fine-tuning. For example, for the real world tasks, you collected 200 trajectories. It would be useful to see how the performance changes as we change the size of the dataset.
5. I am not sure that this method would generalize and I suspect that this might harm the robustness of the pretrained policy. Since you are finetuning offline, it is clear that the REMAC would do well on tasks/environments seen during finetuning – the model knows what the target expert actions are and the model knows what the pretrained policy outputs at these states in the offline dataset. This raises the question of whether or not this fine-tuned model would reliably adapt the pretrained policy’s behavior in unseen states. For example, one issue that might arise is that, at unseen states, the policy might be more uncertain leading to higher entropy of actions from the pretrained policy, and as such your finetuned policy needs to learn how to modify a large number of actions to match the optimal one (which the model does not know yet since this is at test time). Since you are already testing this on $\pi_0$ which demonstrates some generalization, it would be good to evaluate on some tasks/environments not seen during fine-tuning.

[1] Kevin Black, Manuel Y. Galliker, Sergey Levine. Real-Time Execution of Action Chunking Flow Policies.
[2] Yuejiang Liu, Jubayer Ibn Hamid, Annie Xie, Yoonho Lee, Maximilian Du, Chelsea Finn. Bidirectional Decoding: Improving Action Chunking via Guided Test-Time Sampling

**Questions:**

See weaknesses. For me, the most important questions are with regards to clarity and handling latency (see points 1-3 in Weaknesses section).

---

> ### Author Response · Authors · 2025-11-20
> **Rebuttal by Authors (1)**
>
> Thank you for the insightful review! We sincerely appreciate the reviewer’s constructive suggestions and have revised the manuscript to clarify key points and avoid potential misunderstandings. We additionally conducted real-world experiments to analyze REMAC’s performance in low-data regimes and to examine its generalization behavior. We hope these new clarifications and results address your concerns, and we are happy to discuss any further questions.
>
> > ### W1. Clarifying inter-chunk discontinuity and intra-chunk inconsistency.
>
> We thank the reviewer for pointing out sources of confusion. We clarify the concepts as below.
>
> **Inter-chunk discontinuity.** Our definition does not imply that two valid trajectories should produce identical future actions. Instead, inter-chunk discontinuity describes the lack of temporal coherence that can arise when the robot switches from one predicted chunk to the next.
>
> Specifically, flow-matching policies are trained on demonstrations that contain many valid expert trajectories, often with diverse micro-behaviors (e.g., there are many equally correct ways to reach or grasp an object). Since an action chunk contains only a local segment of a trajectory, two consecutive chunks produced at time $t$ and time $t+h$ may correspond to different latent expert modes: chunk at $t$ aligns with expert trajectory $\mathcal T_1$, and chunk at $t+h$ aligns with a different expert trajectory $\mathcal T_2$. Both $\mathcal T_1$ and $\mathcal T_2$ are correct, but the transition between them may introduce a large jump in the action sequence, which we refer to as inter-chunk discontinuity. When there is low/none discontinuity, the policy generates chunks that lie on a consistent motion pattern, producing smooth behavior when consecutive chunks are stitched together.
>
> **Intra-chunk inconsistency.** We assume the policy is competent and that the predicted chunk $\mathbf{A}_t$ is an “optimal” plan for observation ​$\mathbf{o}_t$.
>
> Under a delay of $d$ steps, the robot cannot execute the first $d$ actions of the predicted chunk.
> Instead, it must execute $d$ actions taken from the previous action chunk predicted at timestep $t-h$.
> Once the environment and robot state changes between $t-h$ and $t$, these inherited prefix actions become suboptimal for the current state, because they were conditioned on $\mathbf{o}_{t-h}$, not $\mathbf{o}_t$​. This mismatch between the current observation (perception) and the actually executed prefix  (action) is defined as intra-chunk inconsistency.​
>
> We have added clarifications to the revised manuscript to ensure the definitions are precise and avoid potential misunderstandings.
>
> > ### W2. Clarifying the method, notation, and flow-matching preliminaries.
>
> We thank the reviewer for highlighting these clarity issues. We have revised the manuscript accordingly.
>
> First, we have reintroduced a ***Flow-Matching Policies*** subsection (previously removed) in Sec. 3, which now explicitly defines the predicted flow, the expert flow, and the corresponding training and sampling procedures. We also clarify, in context, which policy each flow is sampled from to avoid any ambiguity in notation.
>
> Second, regarding the notation $\mathbf{x}_{\mathrm{p}}$: in Sec. 4.2, we originally used $\mathbf{x}$​ to denote intermediate action states produced during the iterative flow-matching sampling process, in contrast to the final predicted actions $\mathbf{A}$. We realized that this distinction added unnecessary cognitive burden. In the revised manuscript, we standardize the notation to $\mathbf{A}_t^{\tau}$, introduced in the *Flow-Matching Policies* subsection, to consistently denote the sampled action state at integration step $\tau$. This creates a clear connection between intermediate samples and the resulting action chunk.
>
> Finally, we refined Sec. 4.2 for readability. We remove the abstract function $f$, since it simply denotes a numerical integration update (Euler, Heun, etc.), and instead write out the explicit update rule. Additional explanations have also been added to improve clarity and ease of understanding.

---

> ### Author Response · Authors · 2025-11-20
> **Rebuttal by Authors (2)**
>
> > ### W3. Relationship to RTC and whether REMAC handles latency.
>
> We thank the reviewer for raising this point. Both REMAC and RTC [1] operate under the asynchronous-inference setting (Figure 1(b)), where inference latency is unavoidable and neither method attempts to reduce this latency (which is orthogonal to techniques such as distillation, pruning, or quantization). Instead, both aim to compensate for the performance degradation caused by delayed execution.
>
> The key difference lies in *when* the compensation occurs. **RTC** is a *test-time* method: it performs gradient-based inpainting on the predicted action chunk to enforce inter-chunk temporal continuity. While effective, this introduces nontrivial computational overhead during rollout, creating a direct trade-off between improved consistency and increased inference latency.
> **REMAC**, in contrast, performs *training-time* adaptation. Through masked action chunking and prefix-preserved sampling, the model learns to anticipate and correct both inter-chunk discontinuity and intra-chunk inconsistency (which we newly identify), without adding any extra computation at inference time. As a result, REMAC can handle any valid delay within the chunk horizon while maintaining lower computational latency than RTC during execution. Empirically, REMAC achieves stronger performance and smoother trajectories under the same delay budget.
>
> Finally, the two approaches are complementary rather than conflicting: combining REMAC with RTC (Table 2) yields further improvements, indicating that REMAC enhances robustness while remaining compatible with existing test-time correction strategies.
>
> [1] Real-Time Execution of Action Chunking Flow Policies. NeurIPS 2025.
>
> > ### W4. Data efficiency of fine-tuning.
>
> We appreciate the reviewer’s question regarding data requirements. To evaluate sensitivity to dataset size, we conducted additional real-world experiments using only 10 demonstrations per task, compared to the total 200 demonstrations used in the main paper.
> The average completion progress (10 evaluations per task) is shown below:
> | #Traj | Grasp-Easy | Grasp-Medium | Grasp-Hard |
> |----------|------------|--------------|-------------|
> | 200 | 0.910 | 0.936 | 0.820 |
> | 30 | 0.900| 0.905 | 0.810 |
>
> Even with much less data, REMAC maintains performance very close to the 200-trajectory model, with only mild degradation. These results indicate that REMAC is not data-hungry and remains effective in low-data regimes, thanks to its self-supervised masking mechanism that exposes the model to diverse prefix deviations without requiring additional demonstrations.
>
> > W5. Does REMAC harm the pre-trained policy’s generalization to unseen states?
>
> We thank the reviewer for raising this important concern. REMAC can preserve the pre-trained policy’s generalization while improving its robustness to delay.
>
> To clarify our setup, we first fine-tune $\pi_{0}$ using our collected demonstrations, following the official codebase (both the VLM backbone and action expert are updated). This fine-tuned $\pi_{0}$ serves as the baseline model for all methods, including Naive, RTC, BID, and REMAC. REMAC then only applies LoRA to the baseline model’s action expert, leaving the entire VLM backbone frozen. Thus, REMAC only performs low-rank adjustments on top of the baseline action expert, rather than altering the grounding or perception capabilities of the VLM. This helps to not influence the generalizability of the fine-tuned baseline model.
>
> For empirical justification, we evaluate under two settings: (i) scene/background variation and (ii) novel language prompts. For background variation, we remove the curtains, change the table covering, and introduce additional distractors. REMAC behaves similarly to the baseline model, and both policies consistently ground objects seen during fine-tuning even when distractors or noisy backgrounds are introduced. However, when given unseen language instructions, both the baseline and REMAC struggle with novel tasks or objects (e.g., “cup,” “box”), indicating that this limitation arises from the underlying fine-tuned policy rather than from REMAC.
>
> We further evaluate models trained on $\pi_{0.5}$​ [2], which was found to exhibit stronger open-world generalization in our setting. Under unseen language prompts, the fine-tuned $\pi_{0.5}$​ successfully recognizes unseen objects but still struggles to complete unseen tasks. Applying REMAC on top of it​ preserves this OOD capability and does not introduce noticeable degradation, confirming that our fine-tuning procedure does not hinder language or object generalization beyond what is already present in the base VLA model.
>
> In summary, REMAC inherits the generalization ability of the base model and introduces no additional degradation, since only the action-expert is updated and the vision-language backbone remains unchanged.
>
> [2] $\pi_{0.5}$: a Vision-Language-Action Model with Open-World Generalization. CoRL 2025.

---

> ### Comment · Reviewer_LpbR · 2025-11-23
> **Response to Authors**
>
> Thank you for the detailed response. I will briefly comment on the response from the authors on each of the weaknesses mentioned in the review above.
>
> On W1: the explanation of inter-chunk discontinuity makes sense and is in line with analysis of consistency in existing literature (e.g. in the BID paper). I appreciate the revision to the manuscript which reads more clearly now. One minor comment - the sentence "Suppose $v_\pi(A_t \mid o_t)$ is perfectly trained" in line 174 is a bit too informal; perhaps formalizing/rewording this would be helpful. On W2: I think this was an extremely important improvement to the paper - the readability is significantly improved and, indeed, the cognitive burden on the reader has been decreased. On W3: I found this to be extremely clear and helpful as well. I would *strongly recommend* making this distinction clear in the actual manuscript as well to clarify what the contribution is and exactly in what aspects the reader should expect REMAC to improve on RTC.
>
> I find the authors' response to W4 convincing as well.
>
> My most important concern with the paper was the lack of clarity. I find this work to be quite important and I think the one aspect in which the manuscript could be improved is in clarifying what exactly it is contributing that RTC cannot contribute, and clarifying other concerns the reader might have (e.g. how data efficient the method is, how it affects the base model's generalizability, etc.). I am happy to increase my score.

---

> > ### Author Response · Authors · 2025-11-23
> > **Thank you for your constructive feedback**
> >
> > Thank you very much for your encouraging feedback and recognition of our work! We are glad that our revisions helped clarify the paper and addressed your concerns. Based on your suggestions, we have updated the manuscript in the following ways:
> >
> > - **Line 174 wording refined**: We revised the sentence to the more formal phrasing: “Assume the policy $\mathbf v_{\pi}(\mathbf A_t | \mathbf o_t)$ perfectly captures the underlying environment dynamics and therefore yields the optimal action sequence.”
> >
> > - **Clarified distinction between REMAC and RTC**: We expanded the Related Work section to more clearly articulate how REMAC differs from RTC and what advantages REMAC uniquely provides.
> >
> > - **Added analysis on data efficiency and generalizability**: Appendix Sec. E now includes additional experiments and discussion on REMAC’s behavior in low-data regimes and its effect on the base model’s generalization.
> >
> > Once again, thank you for the time and effort invested in reviewing our work. We sincerely appreciate your constructive insights and are willing to discuss any further questions.

---

### Official Review · Reviewer_NdJX · 2025-10-27

**Soundness:** 3
**Presentation:** 3
**Contribution:** 3
**Rating:** 6
**Confidence:** 3

**Summary:**

The paper proposes REMAC for asynchronous inference in real-world robotics, handling chunk discontinuities by finetuning learned policies via flow matching. The key is to condition the policy on a ground-truth action prefix during training and use flow matching to predict the remaining actions. By randomly sampling delay d and adjusting the flow matching curriculum, REMAC trains the policy to adapt to varying inference delays. Experiments on both simulation and real-world benchmarks show that REMAC outperforms baselines under high inference delay.

**Strengths:**

1. REMAC can adapt to various inference delays d with a single training process, without needing to retrain for each delay. The use of LoRA modules maintains model performance while reducing training overhead.
2. Flow matching enables the model to learn fine-grained continuity between action prefixes and optimal future actions, capturing the dependency between earlier and later actions.
3. REMAC achieves strong results across both simulated and real-world benchmarks.

**Weaknesses:**

1. REMAC has structural requirements on the dataset, which needs to contain diverse future action sequences from the same observation. With this requirement, the model can learn to correct deviating action prefixes resulting from inference delay. Without local adjustment samples in the dataset, REMAC may perform no better than behavioral cloning.
2. During inference, the agent must know how long inference takes in order to select an appropriate prefix length d from the previous action chunk. In real-world settings with variable communication delays, accurately estimating this delay may be challenging.
3. REMAC improves policies' robustness to inference delays only for flow-matching-based policies. If the bottleneck policy is not trained with flow matching, REMAC requires training from scratch.

**Questions:**

1. I'm confused why REMAC outperforms the pretrained policy (Naive) even when d = 0, according to Figure 2. Is it because REMAC uses additional expert data compared with Naive? If the baselines (Naive, RTC, BID) also access the additional expert data, will they outperform REMAC?
2. Can we apply the same dataset used in the pretrained policy (i.e., Naive) to REMAC, so that REMAC does not use additional expert data?
3. Can you further explain the Residual Alignment loss Eq. 4? What's the difference between Eq. 4 and the Prefix Masking loss Eq. 2?
4. In the simualtion tasks, how is the expert dataset D used for REMAC constructed?

---

> ### Author Response · Authors · 2025-11-20
> **Rebuttal by Authors (1)**
>
> Thank you for the insightful review! We have added new experiments and conducted additional analyses to demonstrate REMAC’s robustness to inaccurate delay estimates and its generalizability across different policy classes. We hope the new results and clarifications address your concerns, and we are happy to discuss any further questions.
>
> > ### W1. Dataset requirements and the need for diverse future trajectories.
>
> We thank the reviewer for raising this point. REMAC does **not** require the dataset to contain multiple divergent future trajectories for the same observation. Its corrective ability is learned through a self-supervised temporal masking mechanism: we artificially mask the delay-dependent prefix of each demonstration and train the model to realign the remaining suffix. This synthetic masking creates prefix deviations *within a single trajectory segment*, exposing the model to a wide range of misaligned prefixes without needing multiple future branches. The model therefore learns to infer the correct suffix from the observation under varying delays, rather than memorizing alternative futures. As a result, REMAC does not rely on any special dataset structure beyond standard sequential demonstrations, and remains strictly more robust than behavioral cloning.
>
> Empirically, we validate this in the real-world setting: each task is trained with only 10 demonstrations, where repeated observations across trajectories are extremely rare. Yet REMAC achieves performance comparable to the version trained on 200 demonstrations and still surpasses baselines trained via behavioral cloning. This confirms that REMAC does not depend on trajectory diversity to learn delay-aware corrections.
>
> > ### W2. Difficulty of estimating delay under variable real-world communication latency.
>
> We appreciate the reviewer’s concern. REMAC does not rely on accurate delay estimates to perform well in practice. All real-world results in the paper are obtained under inherently noisy delay measurements, yet REMAC consistently outperforms baselines (Table 3, Figure 4 in main paper). The reasons are as follows.
>
> In real-world deployment, inference delay is naturally noisy due to computation, communication, and sensing jitter. Moreover, because the delay measurement always includes the future VLA inference time, the value fed into the policy is necessarily an estimate rather than ground truth. Following RTC [1], we adopt the same delay estimation strategy: (1) measure the time between receiving the latest observation and receiving the next predicted action chunk on the robot client; (2) convert this latency into a discrete delay value using the controller frequency (Sec. 3); (3) maintain a short queue of recent delay measurements and use the maximum value as the estimate. This procedure intentionally yields a conservative (upper-biased) estimate and effectively handles natural variability.
>
> To stress-test robustness, we further evaluated REMAC under *extremely adversarial delay fluctuations* by corrupting the delay at each inference step with a uniformly sampled value from $[d-1, d, d+1, d+2]$. Even in this harsh regime, REMAC incurs only 1–2 additional failures out of 10 episodes, and these failures occur primarily because persistent overestimation (i.e., repeated $d+2$) causes the robot to pause and exceed the 300-step time limit—not because the policy produces incorrect actions. This indicates that the underlying policy remains stable even under highly unstable delay patterns well beyond what is observed in practice. Overall, REMAC is robust to inaccurate, noisy, and rapidly fluctuating delay estimates, and does not require precise latency measurement to maintain strong performance.
>
> [1] Real-Time Execution of Action Chunking Flow Policies. NeurIPS 2025.

---

> ### Author Response · Authors · 2025-11-20
> **Rebuttal by Authors (2)**
>
> > ### W3. Limitation to flow-matching-based policies.
>
> We thank the reviewer for raising this point. REMAC is not tied to flow-matching policies. The method operates at the action‐chunk level via delay-conditioned masking, and therefore applies to any action chunking policies.
>
> While diffusion policies are a natural extension due to their structural similarity to flow matching, we further show that REMAC also works with **Transformer-based chunking policies**. We adapted REMAC to ACT [2] by applying LoRA on its decoder layers and the action head, and replacing the $\ell_2$ losses in Eq. (3) and (5) with ACT’s $\ell_1$ and KL losses. Below we report results (*success rate / average return*) on two bimanual ACT tasks under different delays, showing that REMAC consistently outperforms both the naive asynchronous baseline and the LoRA-only baseline under different delays, demonstrating that REMAC generalizes beyond flow matching and can be integrated into different action-chunking approaches.
> #### **Transfer Cube (h = 12)**
> | Delay (d) | Naive | +LoRA | Ours |
> |---|-------|--------|--------|
> | 4 | 0.40 / 354.44 &nbsp;&nbsp;&nbsp; | 0.52 / 335.04 &nbsp;&nbsp;&nbsp; | **0.74 / 486.78** |
> | 6 | 0.48 / 348.76 | 0.58 / 352.24 | **0.72 / 508.28** |
> | 8 | 0.46 / 309.32 | 0.68 / 422.62 | **0.68 / 460.86** |
>
> #### **Insert Box (h = 30)**
> |Delay (d)|Naive|+LoRA|Ours|
> |---|--------|---------|---------|
> |0|0.14 / 230.74 &nbsp;&nbsp;&nbsp; |0.20 / 218.74 &nbsp;&nbsp;&nbsp; |**0.18 / 245.72**|
> |5|0.14 / 219.78|0.12 / 179.90|**0.18 / 217.12**|
> |10|0.14 / 216.98|0.10 / 183.94|**0.16 / 220.68**|
>
> *Values shown as: success rate / average return*
>
> [2] Learning Fine-Grained Bimanual Manipulation with Low-Cost Hardware. RSS 2023.
>
> > ### Q1&Q2. Why does REMAC outperform the pretrained policy at d = 0? Does REMAC use additional expert data?
>
> We thank the reviewer for this question. REMAC **does not use any additional expert data**. All methods—including Naive, RTC, BID, and REMAC—are trained on the exact same dataset and have access to the same demonstrations.
>
> Thus, REMAC’s improvement at $d=0$ is **not** due to extra supervision. The gains arise because the masked-chunking strategy produces a more robust policy that mitigates error accumulation during rollout. By fine-tuning with masked prefixes, the policy learns stronger temporal coherence and improved inter-chunk continuity, leading to fewer compounding errors and higher success rates even without delay. Additionally, Table 1 in the main paper shows that simply adding LoRA to the Naive baseline and training with its original objective does not improve performance (and can even degrade it). This confirms that REMAC’s improvement comes from its design rather than increased model capacity.
>
> > ### Q3. Further explanation of Residual Alignment loss.
>
> We thank the reviewer for the insightful question. While the expressions of  $L_\Delta$ and $L_m$ look algebraically similar, they are **not optimization-equivalent**. $L_m$ supervises the absolute actions, pushing the fine-tuned policy toward the expert target. In contrast, $L_\Delta$ explicitly supervises the residual between the backbone output $\mathbf{\tilde u_\tau}$ and the expert action $\mathbf{u_\tau}$, encouraging the model to learn local corrections on top of the frozen pretrained backbone. This residual formulation has several practical effects: (1) It serves as a regularization term, where the LoRA layers are encouraged to learn local adjustment relative to the original policy $\mathbf{v}_\pi$, preventing the fine-tuning process to drift away too far and change policy behavior significantly. (2) Predicting residuals is easier than reconstructing full actions, leading to smoother and more stable optimization.
>
> Empirically, adding $L_\Delta$​ consistently improves performance (Table 1 in main paper), whereas we find simply changing $\lambda_\mathrm{m}$​ to $\lambda_\mathrm{m} + \lambda_{\Delta}$ (Eq. (6)) does make a difference in performance. This indicates the benefit of the residual alignment, and proves that the two terms have different purposes​ during training.
>
> > ### Q4. How is the expert dataset used for REMAC constructed in the simulation tasks?
>
> We follow the dataset construction procedure from RTC. Specifically, we first train multiple expert policies using RPO [3] with different random seeds to obtain diverse but high-quality expert behaviors. We then roll out these expert policies in the environment to collect approximately 1 million environment steps, forming the demonstration dataset.
> This dataset is used invariantly for all methods evaluated in the paper—including Naive, RTC, BID, and REMAC. REMAC does not use any additional demonstrations or privileged data during training and relies solely on this shared expert dataset.
>
> [3] Robust policy optimization in deep reinforcement learning. ArXiv 2022.

---

> > ### Comment · Reviewer_NdJX · 2025-11-24
> >
> > Thank you for your response and additional experiments! I updated my score accordingly.
> > The authors' response resolves my concerns on the additional data usage and dataset structure requirements. I am particularly impressed by your experiments with Transformer-based chunking policies. I believe they are more common than flow-matching policies, and I hope you can add these results to your revised manuscript.

---

> > > ### Author Response · Authors · 2025-11-24
> > > **Thank you for your insightful feedback**
> > >
> > > Thank you very much for your encouraging feedback and recognition of our work! We are delighted that our revisions addressed your concerns. As suggested, we have updated the manuscript by adding a discussion of REMAC’s applicability to Transformer-based chunking policies in **Sec. 5.1 (Ablations)** and providing further details in **Appendix Sec. E.5**.
> > >
> > > Once again, we sincerely appreciate the time and effort you invested in reviewing our paper. Your comments were extremely helpful, and we would be happy to discuss any further questions.

---

### Official Review · Reviewer_n9Cq · 2025-11-01

**Soundness:** 4
**Presentation:** 3
**Contribution:** 4
**Rating:** 8
**Confidence:** 3

**Summary:**

This paper studies real-time robot control with chunk-based VLA policies under asynchronous inference, where inference latency causes the robot to execute stale actions while new ones are being generated. The authors highlight a neglected failure mode—intra-chunk inconsistency—in addition to the known inter-chunk discontinuity issue. They propose REMAC, a training-time strategy that applies prefix masking to simulate partial stale-action execution, a self-conditioned curriculum to mitigate exposure bias, and residual LoRA finetuning to correct the base policy. The method increases robustness under varying latencies without adding inference overhead. Experiments on Kinetix and a real Franka arm show improved success rates and smoother execution across delay conditions.

**Strengths:**

- Identifies and formalizes intra-chunk inconsistency as a distinct source of degradation in asynchronous execution.
- Simple and computationally efficient method that is compatible with existing VLA architectures.
- Strong empirical evidence across simulation and real-robot settings, including latency sweeps and ablation studies.
- No additional inference latency, in contrast to recent test-time smoothing approaches (e.g., RTC).
- Parameter-efficient finetuning strategy that preserves the backbone’s capability.
- Compatible with BID/RTC test-time methods, demonstrating good composability with existing techniques.

**Weaknesses:**

- The method assumes access to accurate and bounded delay estimates. It is unclear how robust the approach is when latency measurements are noisy, rapidly fluctuating, or adversarially spiky.
- Training samples delays uniformly, but real-world latency tends to be bursty and temporally correlated. Additional evaluation under realistic network- and compute-induced delay profiles would strengthen the claims.
- Finetuning with masked actions may shift the behavior of the underlying VLA model. The paper does not report out-of-distribution or language generalization results, making it difficult to assess whether broader generalization and grounding capabilities are preserved.
- The method is demonstrated only on flow-matching VLA architectures. Since REMAC integrates with the sampling process and residual flow fields, it is unclear whether the approach directly applies to non-flow controllers (e.g., autoregressive VLA policies or transformer-based continuous action models). A discussion or preliminary evidence on generality across policy parameterizations would strengthen the paper.

**Questions:**

1. How robust is REMAC to inaccurate delay estimates or rapidly varying latency?
2. Does delay-aware finetuning impact generalization to unseen tasks/objects or language inputs?
3. Would a continuous or learned delay embedding outperform discrete integer conditioning?
4. How does the method behave when real latency exceeds the maximum trained delay?
5. Can the approach be extended to handle coupled observation latency, not just action delay?
6. The method uses a fixed chunk length, but in practice the optimal horizon may vary with latency and task demands. Can the approach be extended to dynamically adjust chunk length based on system delay or task complexity?

---

> ### Author Response · Authors · 2025-11-20
> **Rebuttal by Authors (1)**
>
> Thank you for the insightful review! Your comments have helped us further analyze REMAC’s behavior under different conditions, especially under varying latency. In response, we conducted additional ablations and analyses in both simulation and real-world settings. We hope the new results and clarifications address your concerns, and we are happy to discuss any further questions.
>
> > ### W1, W2 & Q1. Robustness to inaccurate, rapidly fluctuating and spiky delay estimates. Inclusion of  realistic conditions.
>
> We thank the reviewer for raising these important questions. We conducted additional evaluations in both simulation and the real-world to assess REMAC’s robustness under noisy, rapidly fluctuating, and adversarially spiky delay patterns.
> ### **(1) Simulation experiments**
> We fix the true execution delay to $d$, but deliberately pass incorrect delay values to the policy via the following methods:
>  - Noisy and rapidly fluctuating: For each policy call, we sample the delay from $[d-1, d, d+1]$ (66% inaccurate delay estimates)
> - Spiky delays: With 10% probability, we replace the delay with the maximum valid value to simulate infrequent but large latency spikes.
>
> Below we report the average performance over 12 Kinetix tasks:
> | Setting | d=0 | d=1 | d=2 | d=3 | d=4 |
> |--------|-----|-----|-----|-----|-----|
> | RTC | 0.817 &nbsp;&nbsp;&nbsp;| 0.778 | 0.733 | 0.665 | 0.588 |
> | Ours | **0.877** | **0.860** | **0.832** | **0.796** | **0.760** |
> | RTC + Noisy & Fluctuating | 0.816 | 0.770 | 0.728 | 0.653 | 0.573 |
> | Ours + Noisy & Fluctuating | **0.866** | **0.837** | **0.799** | **0.746** | **0.757** |
> | RTC + Noisy & Fluctuating & Spiky | 0.814 | 0.772 | 0.728 | 0.653 | 0.575 |
> | Ours + Noisy & Fluctuating & Spiky &nbsp;&nbsp;&nbsp;| **0.820** | **0.796** &nbsp;&nbsp;&nbsp;| **0.769** &nbsp;&nbsp;&nbsp;| **0.718** &nbsp;&nbsp;&nbsp;| **0.757** &nbsp;&nbsp;&nbsp;|
>
> Under extremely ill-conditioned latency sequences, both RTC and REMAC degrade gracefully—not catastrophically. Importantly, REMAC under corrupted delays still outperforms RTC under accurate delays, demonstrating strong robustness to delay misestimation.
>
> ### **(2) Real-world experiments**
> All of our real-world evaluations already operate under realistic network- and compute-induced delay profiles, where delay measurements are naturally noisy and temporally correlated. The measured delay includes multiple sources of compute-induced latency—VLA inference time, network transmission, file I/O, memory contention, and system-level scheduling jitter. Because the delay measurement always includes inference time, it is necessarily historical. Following RTC [1], we therefore use the maximum delay observed over a recent window as the estimate passed to the policy. Consequently, all real-world results (Table 3 and Figure 4) are obtained under inherently imperfect delay estimates, yet REMAC still performs well and remains robust under these realistic latency conditions.
>
> To further stress-test the system, we artificially corrupted the delay signal by randomly sampling from $[d−1,d,d+1,d+2]$, producing noisy, rapidly fluctuating, and spiky delay inputs. This causes only 1–2 additional failures out of 10 trials, primarily due to occasional overestimation (i.e., receiving $d+2$), which leads the robot to temporarily pause and exceed the 300-step time limit. Importantly, this is a timeout artifact rather than an indication of policy instability.
>
> ### **(3) An inherent enhancement for robustness**
> We would also like to highlight an inherent robustness property of asynchronous execution. The inference delay is defined as $d := \left\lfloor \delta / \Delta t \right\rfloor$, where $\delta$ is the continuous inference latency and $\Delta t$ is the controller sampling period (67ms at 15Hz). For the discrete delay $d$ to fluctuate by ±1 step, the underlying latency must shift by more than 67 ms—an extreme fluctuation rarely observed in practice. This discretization effect naturally smooths noise in $\delta$, making all asynchronous methods, including ours, inherently robust to moderate variations in continuous latency.
>
> [1] Real-Time Execution of Action Chunking Flow Policies. NeurIPS 2025.

---

> ### Author Response · Authors · 2025-11-20
> **Rebuttal by Authors (2)**
>
> > ### W3 & Q2. Effect of delay-aware fine-tuning on VLA generalization.
>
> We thank the reviewer for raising this important concern. Our design of REMAC can help to **preserve** the pretrained policy’s generalization while improving its robustness to delay.
>
> To clarify our setup, we begin with the open-source $\pi_{0}$ model and finetune it using our collected demonstrations, following the official codebase (both the VLM backbone and action expert are updated). This fine-tuned $\pi_{0}$ serves as the baseline model for all methods, including Naive, RTC, BID, and REMAC. REMAC then only applies LoRA to the baseline model’s **action expert**, leaving the entire VLM backbone frozen. Thus, REMAC only performs local, low-rank adjustments on top of the baseline action expert, rather than altering the grounding or perception capabilities of the VLM. This design helps to not influence the generalizability of the finetuned baseline model.
>
> For empirical justification, we evaluate under two settings: (i) scene/background variation and (ii) novel language prompts. For background variation, we remove the curtains, change the table covering, and introduce additional distractors. REMAC behaves similarly to the baseline model, and both policies consistently ground objects seen during fine-tuning even when distractors or noisy backgrounds are introduced. However, when given unseen language instructions, both the baseline and REMAC struggle with novel tasks or objects (e.g., “cup,” “box”), indicating that this limitation arises from the underlying fine-tuned policy rather than from REMAC.
>
> We further evaluate models trained on $\pi_{0.5}$​ [2], which was found to exhibit stronger open-world generalization in our setting. Under unseen language prompts, the fine-tuned $\pi_{0.5}$​ successfully recognizes unseen objects but still struggles to complete unseen tasks. Applying REMAC on top of it​ preserves this OOD capability and does not introduce noticeable degradation, confirming that our fine-tuning procedure does not hinder language or object generalization beyond what is already present in the base VLA model.
>
> In summary, REMAC inherits the generalization ability of the base model and introduces no additional degradation, since only the action-expert is updated and the vision-language backbone remains unchanged.
>
> [2] $\pi_{0.5}$: a Vision-Language-Action Model with Open-World Generalization. CoRL 2025.
>
> > ### W4. Generalizability across different policies.
>
> We thank the reviewer for raising this point. REMAC is not tied to flow-matching policies. The method operates at the action‐chunk level via delay-conditioned masking, and therefore applies to any action chunking policies.
>
> While diffusion policies are a natural extension due to their structural similarity to flow matching, we further show that REMAC also works with **Transformer-based chunking policies**. We adapted REMAC to ACT [3] by applying LoRA on its decoder layers and the action head, and replacing the $\ell_2$ losses in Eq. (3) and (5) with ACT’s $\ell_1$ and KL losses. Below we report results (*success rate / average return*) on two bimanual ACT tasks under different delays, showing that REMAC consistently outperforms both the naive asynchronous baseline and the LoRA-only baseline under different delays, demonstrating that REMAC generalizes beyond flow matching and can be integrated into different action-chunking approaches.
> #### **Transfer Cube (h = 12)**
> | Delay (d) | Naive | +LoRA | Ours |
> |---|-------|--------|--------|
> | 4 | 0.40 / 354.44 &nbsp;&nbsp;&nbsp; | 0.52 / 335.04 &nbsp;&nbsp;&nbsp; | **0.74 / 486.78** |
> | 6 | 0.48 / 348.76 | 0.58 / 352.24 | **0.72 / 508.28** |
> | 8 | 0.46 / 309.32 | 0.68 / 422.62 | **0.68 / 460.86** |
>
> #### **Insert Box (h = 30)**
> |Delay (d)|Naive|+LoRA|Ours|
> |---|--------|---------|---------|
> |0|0.14 / 230.74 &nbsp;&nbsp;&nbsp; |0.20 / 218.74 &nbsp;&nbsp;&nbsp; |**0.18 / 245.72**|
> |5|0.14 / 219.78|0.12 / 179.90|**0.18 / 217.12**|
> |10|0.14 / 216.98|0.10 / 183.94|**0.16 / 220.68**|
>
> *Values shown as: success rate / average return*
>
> [3] Learning Fine-Grained Bimanual Manipulation with Low-Cost Hardware. RSS 2023.

---

> ### Author Response · Authors · 2025-11-20
> **Rebuttal by Authors (3)**
>
> > ### Q3: Would a continuous or learned delay embedding outperform discrete integer conditioning?
>
> We thank the reviewer for the thoughtful question. In practice, the model is not directly conditioned on a raw integer delay $d$. Instead, delay is encoded through the prefix mask $\mathbf{m}_d$ (Eq. (2)), which specifies which portion of the chunk should be treated as “already executed”. This mask is used both during optimization and during sampling, serving as the mechanism through which delay information is conveyed.
>
> In our ablations, we experimented with learning a continuous mask embedding, and found that it brings mixed effects.
> Specifically, we pass $\mathbf{m}_d$ through an extra projection layer inside the LoRA modules, producing a continuous embedding that is fused with the action input. As shown in Figure 10(b) of the revised manuscript, this yields mixed effects: some tasks show slight improvement, while others exhibit no meaningful change. This suggests that introducing a separate continuous embedding does not consistently offer additional benefits.
>
> > ### Q4. Delay values beyond the trained range.
>
> We thank the reviewer for raising this point. The delay range used during training does not limit the delays that REMAC can handle at test time, and we recognize that our original notation ($d_\text{max}$, $d_\text{min}$) might have misled readers. To prevent this misunderstanding, we have renamed these parameters to $⁡q_\text{max}$ and $q_\text{min}$ in the revised manuscript (Sec 4.3), emphasizing that they control only the masking strength during training, not the range of inference delays the policy can accommodate. And as $q$ is annealed from $⁡q_\text{max}$ to $q_\text{min}$, the model is gradually exposed to longer valid suffixes, forming a curriculum that stabilizes training. Importantly, this process does not impose any limitation on the inference-time delay.
>
> To clarify this empirically, we added an ablation (**Figure 10(a)**) in the revised manuscript showing that REMAC is robust across different $(⁡q_\text{min}, ⁡q_\text{max})$ configurations. Even under $⁡q_\text{max}=0,2$ and $⁡q_\text{min}=2,4$, the learned policy performs well under delays larger than $⁡q_\text{max}$ and delays smaller than $q_\text{min}$, confirming that they do not bound the delays REMAC can handle. This is expected: because at inference time the delay enters through the binary mask $\mathbf{m}_{d}$, which cleanly partitions the chunk into a prefix that is *copied from previously predicted chunk* and a suffix that is *newly generated*. This mechanism remains viable for any valid delay $d$ within the rollout window.
>
> The only limit on usable delay is the mechanical constraint $d \leq P-h$ ($P$ is the prediction horizon and $h$ is the execution horizon). When the delay exceeds the number of yet-unexecuted actions in the previous chunk, the robot simply runs out of queued actions and pauses, and performance will vary based on task progress and the learned policy’s implicit dynamics. This limitation is inherent to all action-chunking policies under asynchronous execution, rather than specific to REMAC.
>
> > ### Q6. Can the approach be extended to dynamically adjust chunk length based on system delay or task complexity?
>
> We appreciate the reviewer’s thoughtful question. Dynamically adapting the executed chunk length is indeed an interesting direction, and it remains an open problem in both synchronous and asynchronous action chunking frameworks. Existing analyses (e.g., [4]) show that even in synchronous settings, the relationship between chunk length, task demands, and policy performance is highly non-monotonic: shorter horizons improve reactivity but reduce stability, while longer horizons improve temporal consistency at the cost of responsiveness. They inferred that a good horizon is related to both the environment dynamics and the quality of the learned policy. In asynchronous settings, the problem only becomes even more complex due to the involvement of additional execution errors.
>
> Empirically, we also observe a mixed trend: under the same delay, some tasks benefit from larger horizons, others favor smaller ones, and some exhibit non-monotonic behavior. While for the same task, different policies have different optimal horizons. To the best of our knowledge, no prior work has proposed a reliable mechanism for dynamically adjusting chunk length at test time. Therefore, we conclude that horizon determination is a complex and underexplored area of research, and is hard to be heuristically determined via system delay or task complexity.
>
> We view this as an important long-term research direction. However, it lies beyond the scope of the present paper, whose goal is to improve fixed-horizon chunking policies under delay. We will add a discussion of this point to the revised manuscript.
>
> [4] Bidirectional Decoding: Improving Action Chunking via Guided Test-Time Sampling. ICLR 2025.

---

> ### Author Response · Authors · 2025-11-20
> **Rebuttal by Authors (4)**
>
> > ### Q5. Can the approach be extended to handle coupled observation latency, not just action delay?
>
> We thank the reviewer for raising this important point. In the current implementation, we follow prior work and model only action (inference) delay, which is measured on the robot client side as the time between receiving observations and receiving the predicted action chunk. For fair comparison with RTC and other baselines, observation latency was not modeled in the main paper.
>
> However, we analyze that REMAC can naturally handle observation latency without requiring any architectural modifications. Conceptually, both observation delay and action delay lead to the same fundamental challenge: perception–action misalignment. In both cases, the environment may change during the latency window, causing the robot to execute actions that correspond to a stale observation. Therefore, because REMAC’s mechanism is delay-aware rather than modality-specific, handling observation latency requires only a simple adjustment: when both types of delays are present, we can replace the delay input by $d_{a} \leftarrow d_{a} + d_{o}$, where $d_{o}$ is the observation delay and $d_{a}$ is the action delay. No modification to the model architecture or the training objective is needed.
>
> **Simulation results.** To assess REMAC’s ability to handle observation delay, we evaluate REMAC, RTC, and the naive baseline on the Kinetix benchmark under varying observation delays $d_{o}$ and action delays $d_{a}$​. Observation delay is simulated by feeding a stale observation (from $t-d_{o}$) into the policy during prediction. To isolate each method’s inherent robustness, we do not reveal the existence of observation delay to any method (i.e., all models assume only action delay). We report the average success rate under different observation delays across 12 tasks below.
>
> $d_{o}=1$:
> | $d_a$ | 0 | 1 | 2 | 3 | 4 |
> |--------|-------|-------|-------|-------|-------|
> | Naive &nbsp;&nbsp;&nbsp; | 0.608 &nbsp;&nbsp;&nbsp; | 0.519 &nbsp;&nbsp;&nbsp; | 0.482 &nbsp;&nbsp;&nbsp; | 0.427 &nbsp;&nbsp;&nbsp; | 0.394 |
> | RTC   | 0.628 | 0.590 | 0.568 | 0.532 | 0.492 |
> | Ours  | **0.676** | **0.675** | **0.714** | **0.746** | **0.690** |
>
> $d_{o}=2$:
>  $d_a$ | 0 | 1 | 2 | 3 | 4 |
> |--------|-------|-------|-------|-------|-------|
> | Naive &nbsp;&nbsp;&nbsp;| 0.407 &nbsp;&nbsp;&nbsp;| 0.357 &nbsp;&nbsp;&nbsp;| 0.354 &nbsp;&nbsp;&nbsp;| 0.329 &nbsp;&nbsp;&nbsp;| 0.335 |
> | RTC   | 0.419 | 0.404 | 0.403 | 0.388 | 0.382 |
> | Ours  | **0.462** | **0.465** | **0.520** | **0.538** | **0.593** |
>
> The results show that: (1) Observation delay degrades performance for all methods, as expected. (2) REMAC consistently outperforms both RTC and the naive baseline across all $(d_{o}, d_{a})$ combinations. (3) REMAC is more affected when $d_{a}$ is small,because the influence of observation delay becomes relatively larger; when inference delay is high, the marginal impact of $d_{o}$​ is reduced. These findings validate that observation delay and action delay induce the same core failure mode and demonstrate that REMAC can effectively mitigate both.
>
> **Real-world analysis.** Our real-world system already exhibits non-zero observation latency due to camera processing and communication overhead, and REMAC performs well under these conditions without any explicit modeling of $d_{o}$. To explicitly condition on observation delay and apply targeted correction, additional system instrumentation—such as sensor timestamping and clock synchronization between the perception pipeline and robot controller—would be required to obtain accurate delay estimates. To remain consistent with prior work, we currently focus specifically on inference (action). We will clarify this design choice in the revised manuscript and plan to incorporate explicit observation-delay modeling in future system versions.

---

> > ### Comment · Reviewer_n9Cq · 2025-11-25
> >
> > The authors’ rebuttal is thorough and well-supported, offering new experiments and clear explanations that address all of my earlier concerns. The robustness analyses under noisy, fluctuating, and spiky delays significantly strengthen the empirical claims. Extra demonstration on REMAC extends beyond flow-matching to transformer-based architectures like ACT further supports the generalizability to any VLA models. The discussion on handling observation latency, delay ranges, and the inherent difficulty of dynamic horizon selection is reasonable and aligns with the core mechanism. Overall, the rebuttal substantially improves the submission, and I'm satisfied with the authors’ responses.

---

> > > ### Author Response · Authors · 2025-11-25
> > > **Thank you for your constructive feedback**
> > >
> > > Thank you very much for your encouraging feedback and recognition of our work! We are glad that the rebuttal helped address your concerns. We have updated the manuscript to include analyses under noisy, fluctuating, and spiky delays, as well as an extension to Transformer-based architectures such as ACT.
> > >
> > > Once again, thank you for the time and effort invested in reviewing our work. We sincerely appreciate your insightful comments and are happy to discuss any further questions.

---

### Official Review · Reviewer_szGQ · 2025-11-04

**Soundness:** 3
**Presentation:** 3
**Contribution:** 2
**Rating:** 6
**Confidence:** 3

**Summary:**

This paper addresses real-time robot execution with action chunks. The authors identify _intra-chunk inconsistency_ as a previously overlooked issue. They propose a method for learning a delay-aware policy by masking the portion of the prefix that restricts supervision to only the executable slice of each chunk. The proposed method incurs no additional latency.

**Strengths:**

- The paper makes solid contributions to an important practical problem. Asynchronous inference is a natural system-level solution for real-time execution. Identifying intra-chunk inconsistency as a distinct failure mode adds value.
- The paper does a thorough experimental analysis, comparing with the state-of-the-art method RTC, and doing ablation studies on each component of the proposed method.
- Figure 1(c) provides a great concrete example of the problem the paper is aiming to solve.

**Weaknesses:**

- The paper cites intra-chunk inconsistency as a core motivation for the proposed method, but doesn't provide any direct evidence that this is a major issue.
- Is there a way to apply this idea to policy classes other than flow-matching?
- I don't understand the residual alignment term. Don't the two $\tilde{u}$ terms in eq (4) cancel out, making it equivalent to (2)? Is there a typo somewhere, or am I missing something?
- The method requires specifying d_max as a hyperparameter during training. Do you have a sense of what happens when the model encounters delays longer than d_max? Does performance degrade gracefully or fail catastrophically?

**Questions:**

Please see the weaknesses section above.

---

> ### Author Response · Authors · 2025-11-20
> **Rebuttal by Authors (1)**
>
> Thank you for the insightful review! We have added new ablation experiments, conducted additional analyses, and revised the manuscript to clarify the intended methodology and avoid potential misunderstandings. We hope our responses adequately address your concerns, and we are happy to discuss any further questions.
> > ### Q1. Evidence for intra-chunk inconsistency being a major issue.
>
> We thank the reviewer for this important question. Intra-chunk inconsistency occurs when, given an input observation at timestep $t$, the robot first executes actions from an action chunk that was based on an older observation, rather than from the newly generated chunk aligned with the current observation. As a result, the first few executed actions no longer correspond to the perceptual context at timestep $t$. This misalignment becomes especially harmful near behavior-change points—for example, when the VLA predicts a new plan based on the latest observation, but the robot is still executing the stale prefix from the previous chunk.
>
> We also empirically show that intra-chunk inconsistency is not only present but closely relevant to performance degradation.
> We quantify this inconsistency using the cosine distance between *the predicted chunk* and *the actually executed prefix*. The table below reports *cosine distance / success rate* under varying delays for different tasks. Across all tasks, divergence grows monotonically with delay, and performance drops in parallel (the same trend holds for the $\ell_2$ distance). This quantitative trend indicates that intra-chunk inconsistency is correlated with the failure of chunked policies under asynchronous inference. Under REMAC, we find that the divergence between predicted chunk and executed prefix approaches 0, implying the effectiveness of our method in alleviating intra-chunk inconsistency.
> | Delay \(d\) | Car Launch | Catapult | Grasp Easy | Lunar Lander | Mjc Swimmer |
> |-------------|------------|----------|-------------|---------------|--------------|
> | **0** | 0.00 / 0.67 | 0.00 / 0.63 | 0.00 / 0.97 | 0.00 / 0.84 | 0.00 / 0.91 |
> | **1** | 0.23 / 0.47 | 0.24 / 0.38 | 0.13 / 0.62 | 0.21 / 0.80 | 0.19 / 0.84 |
> | **2** | 0.37 / 0.40 | 0.38 / 0.38 | 0.22 / 0.64 | 0.35 / 0.69 | 0.34 / 0.64 |
> | **3** | 0.49 / 0.34 | 0.51 / 0.34 | 0.32 / 0.52 | 0.49 / 0.53 | 0.48 / 0.36 |
> | **4** | 0.67 / 0.33 | 0.67 / 0.31 | 0.43 / 0.53 | 0.63 / 0.41 | 0.63 / 0.18 |
>
> *Values shown as: cosine distance / success rate*
> > ### Q2. Applicability beyond flow-matching policies.
>
> We thank the reviewer for raising this point. REMAC is not tied to flow-matching policies. The method operates at the action‐chunk level via delay-conditioned masking, and therefore applies to any action chunking policies.
>
> While diffusion policies are a natural extension due to their structural similarity to flow matching, we further show that REMAC also works with **Transformer-based chunking policies**. We adapted REMAC to ACT [1] by applying LoRA on its decoder layers and the action head, and replacing the $\ell_2$ losses in Eq. (3) and (5) with ACT’s $\ell_1$ and KL losses. Below we report results (*success rate / average return*) on two bimanual ACT tasks under different delays, showing that REMAC consistently outperforms both the naive asynchronous baseline and the LoRA-only baseline under different delays, demonstrating that REMAC generalizes beyond flow matching and can be integrated into different action-chunking approaches.
> #### **Transfer Cube (h = 12)**
> | Delay (d) | Naive | +LoRA | Ours |
> |---|-------|--------|--------|
> | 4 | 0.40 / 354.44 &nbsp;&nbsp;&nbsp; | 0.52 / 335.04 &nbsp;&nbsp;&nbsp; | **0.74 / 486.78** |
> | 6 | 0.48 / 348.76 | 0.58 / 352.24 | **0.72 / 508.28** |
> | 8 | 0.46 / 309.32 | 0.68 / 422.62 | **0.68 / 460.86** |
>
> #### **Insert Box (h = 30)**
> |Delay (d)|Naive|+LoRA|Ours|
> |---|--------|---------|---------|
> |0|0.14 / 230.74 &nbsp;&nbsp;&nbsp; |0.20 / 218.74 &nbsp;&nbsp;&nbsp; |**0.18 / 245.72**|
> |5|0.14 / 219.78|0.12 / 179.90|**0.18 / 217.12**|
> |10|0.14 / 216.98|0.10 / 183.94|**0.16 / 220.68**|
>
> *Values shown as: success rate / average return*
>
> [1] Learning Fine-Grained Bimanual Manipulation with Low-Cost Hardware. RSS 2023.

---

> ### Author Response · Authors · 2025-11-20
> **Rebuttal by Authors (2)**
>
> > ### Q3. Understanding the residual alignment term.
>
> We thank the reviewer for the insightful question. While the expressions of  $L_\Delta$ and $L_m$ look algebraically similar, they are **not optimization-equivalent**. $L_m$ supervises the absolute actions, pushing the fine-tuned policy toward the expert target. In contrast, $L_\Delta$ explicitly supervises the residual between the backbone output $\mathbf{\tilde u_\tau}$ and the expert action $\mathbf{u_\tau}$, encouraging the model to learn local corrections on top of the frozen pretrained backbone. This residual formulation has several practical effects: (1) It serves as a regularization term, where the LoRA layers are encouraged to learn local adjustment relative to the original policy $\mathbf{v}_\pi$, preventing the fine-tuning process to drift away too far and change policy behavior significantly. (2) Predicting residuals is easier than reconstructing full actions, leading to smoother and more stable optimization.
>
> Empirically, adding $L_\Delta$​ consistently improves performance (Table 1 in main paper), whereas we find simply changing $\lambda_\mathrm{m}$​ to $\lambda_\mathrm{m} + \lambda_{\Delta}$ (Eq. (6)) does make a difference in performance. This indicates the benefit of the residual alignment, and proves that the two terms have different purposes​ during training.
>
> > ### Q4. Interpretation of $d_\text{max}$​ and its impact on performance
>
> We thank the reviewer for raising this point and recognize that our original notation misled readers to interpret these as a limitation on the delays REMAC can handle. The original symbols $d_\text{max}$ and $d_\text{min}$ were **not** intended to denote the maximum inference delay supported by the method. To avoid this confusion, we have renamed them to $⁡q_\text{max}$ and $q_\text{min}$ in the revised manuscript (Sec 4.3). These parameters control only the **strength of prefix masking during training**, i.e., how aggressively we corrupt the prefix of the action chunk to simulate the loss of information during rollout to enhance policy robustness. And as $q$ is annealed from $⁡q_\text{max}$ to $q_\text{min}$, the model is gradually exposed to longer valid suffixes, forming a curriculum that stabilizes training. Importantly, this process does not impose any limitation on the inference-time delay.
>
> To clarify this empirically, we added an ablation (**Figure 10(a)**) in the revised manuscript showing that REMAC is robust across different $(⁡q_\text{min}, ⁡q_\text{max})$ configurations. Even under $⁡q_\text{max}=0,2$ and $⁡q_\text{min}=2,4$, the learned policy performs well under delays larger than $⁡q_\text{max}$ and delays smaller than $q_\text{min}$, confirming that they do not bound the delays REMAC can handle. This is expected: because at inference time the delay enters through the binary mask $\mathbf{m}_{d}$, which cleanly partitions the chunk into a prefix that is *copied from previously predicted chunk* and a suffix that is *newly generated*. This mechanism remains viable for any valid delay $d$ within the rollout window.
>
> The only limit on usable delay is the mechanical constraint $d \leq P-h$ ($P$ is the prediction horizon and $h$ is the execution horizon). When the delay exceeds the number of yet-unexecuted actions in the previous chunk, the robot simply runs out of queued actions and pauses, and performance will vary based on task progress and the learned policy’s implicit dynamics. This limitation is inherent to all action-chunking policies under asynchronous execution, rather than specific to REMAC.

---

### Author Response · Authors · 2025-11-28
**Summary of Rebuttal and Manuscript Revisions**

Dear AC and Reviewers,

Thank you for the time and effort devoted to reviewing our work! Since the effective author–reviewer discussion period ended earlier than expected due to the OpenReview information leak on ***Nov 27***, we would like to provide a brief summary of the rebuttal phase.

The paper received an initial rating of **6, 6, 8, 6**, and as of ***Nov 25*** the updated ratings are **8, 8, 8, 6**. We appreciate the active engagement from Reviewers LpbR, NdJX, and n9Cq, and are glad that our rebuttal helped address their concerns, reflected in the rating updates ($6 \rightarrow 8$ on Nov 23, $6 \rightarrow 8$ on Nov 24, and $8 \rightarrow 8$ on Nov 25). We would also be very happy to address any remaining questions from Reviewer szGQ if the opportunity arises.

In response to the reviewers’ constructive feedback, we have revised the manuscript by:
 - Improving clarity of exposition and reducing potential ambiguity to enhance readability.
 - Adding discussion and experiments demonstrating REMAC’s applicability to other policy types, including Transformer-based chunking policies (e.g., ACT).
 - Expanding analyses of REMAC’s robustness under noisy, fluctuating, and spiky delay conditions, as well as its behavior in low-data regimes and its effect on generalization.

Finally, we would like to thank all reviewers for their detailed assessments and insightful feedback, and the AC for facilitating the review process. Their contributions have greatly helped us strengthen the paper!

---

### Meta-Review · Area_Chair_o1dA · 2025-12-28

**Summary:**

The paper proposes a compelling approach to an important problem. The reviewers all agree that the paper merits acceptance.

**Reviewer Concerns:**

Not relevant since I am recommending acceptance.

**Reviewer Scores:**

Not relevant since I am recommending acceptance.

---

### Decision · Program_Chairs · 2026-01-26

Accept (Poster)